# Mechanism of synergistic activation of Arp2/3 complex by cortactin and N-WASP

**Luke A Helgeson[1,2], Brad J Nolen[1,2]\***

[1]Institute of Molecular Biology, University of Oregon, Eugene, United States;
[2]Department of Chemistry and Biochemistry, University of Oregon, Eugene,
United States

**Abstract** Nucleation promoting factors (NPFs) initiate branched actin network assembly by activating Arp2/3 complex, a branched actin filament nucleator. Cellular actin networks contain multiple NPFs, but how they coordinately regulate Arp2/3 complex is unclear. Cortactin is an NPF that activates Arp2/3 complex weakly on its own, but with WASP/N-WASP, another class of NPFs, potently activates. We dissect the mechanism of synergy and propose a model in which cortactin displaces N-WASP from nascent branches as a prerequisite for nucleation. Single-molecule imaging revealed that unlike WASP/N-WASP, cortactin remains bound to junctions during nucleation, and specifically targets junctions with a ~160-fold increased on rate over filament sides. N-WASP must be dimerized for potent synergy, and targeted mutations indicate release of dimeric N-WASP from nascent branches limits nucleation. Mathematical modeling shows cortactin-mediated displacement but not N-WASP recycling or filament recruitment models can explain synergy. Our results provide a molecular basis for coordinate Arp2/3 complex regulation.

## Introduction

Orchestration of many complex cellular processes, including cellular motility, endocytosis, and cytokinesis, requires tight control of the assembly and disassembly of actin filament networks (*Chhabra and Higgs, 2007*; *Pollard and Cooper, 2009*). Actin-related protein (Arp)-2/3 complex is an important actin cytoskeletal regulator that mediates the assembly of branched actin filament networks by nucleating new (daughter) filaments from the sides of pre-existing (mother) filaments (*Figure 1A*) (*Goley and Welch, 2006*; *Rotty et al., 2013*). When isolated from most species, the complex is inactive, and activation requires binding to the side of a preformed actin filament and association with a nucleation promoting factor (NPF) protein (*Figure 1A*) (*Pollard, 2007*; *Achard et al., 2010*). In addition to binding Arp2/3 complex, NPFs discovered to date bind either actin monomers (type I NPFs) or filaments (type II NPFs) (*Goley and Welch, 2006*) (*Figure 1B,C*). Cellular branched actin structures contain multiple NPFs, including representatives from both classes, which frequently have non-redundant roles in actin network assembly (*Galletta et al., 2008*; *Yamaguchi et al., 2005*; *Ayala et al., 2008*). However, the mechanism by which multiple NPFs coordinately regulate Arp2/3 complex activity is poorly understood.

WASP (Wiskot-Aldrich syndrome) and Scar (suppressor of cAR) family proteins, the best studied type I NPFs, have a minimal Arp2/3-activation region called VCA (verprolin homology, central, acidic, *Figure 1B*) (*Goley and Welch, 2006*). The V and C regions of VCA bind actin monomers (*Kelly et al., 2006*), and CA binds two sites on Arp2/3 complex (*Padrick et al., 2011*; *Ti et al., 2011*). Chemical crosslinking assays demonstrate that one CA site is on Arp3 and the other spans Arp2 and ARPC1 (*Padrick et al., 2011*). Using two CA binding sites, VCA is thought to recruit actin monomers to the Arp2 and Arp3 subunits, stimulating a conformational rearrangement to create an Arp2-Arp3-actin hetero-oligomer that mimics a stable actin filament nucleus (*Padrick et al., 2011*; *Hetrick et al., 2013*). Monomeric vs oligomeric VCA regions activate the complex with distinct kinetics, presumably

**\*For correspondence:** bnolen@
uoregon.edu

**Competing interests:** The
authors declare that no
competing interests exist.

**Reviewing editor:** Wesley
Sundquist, University of Utah,
United States

**eLife digest** Cells constantly sense, and react to, their environments. They can monitor or alter their surroundings by taking up or secreting various substances, and may also migrate toward food supplies, or toward signaling molecules—for example, to clot blood or heal wounds. These actions depend on the cytoskeleton, a protein meshwork that gives cells their shape; allows them to transport materials into, out of, or across their cytoplasms; and enables them to move.

The filaments of the cytoskeleton are constructed from several different types of proteins, one of which is called actin. In response to signals, actin can assemble into linear filaments, or can form branches with one end anchored on an existing filament. Branch formation requires the Arp2/3 complex, which initiates and anchors branches on existing filaments, and also various 'nucleation-promoting factors' (NPFs), which turn on the branching activity of the Arp2/3 complex.

Two types of NPFs have been identified: type I interact with individual actin molecules, while type II bind to actin filaments. Previous work has shown that type I NPFs—including the N-WASP protein—have a specialized domain called VCA that binds to both the Arp2/3 complex and to actin molecules. VCA brings actin molecules to the branch site, which initiates branch formation, but how N-WASP collaborates with type II NPFs to build branches is not well understood.

Helgeson and Nolen now examine how a type II NPF called cortactin works with the Arp2/3 complex and N-WASP to construct new branches on actin filaments in vitro. Cortactin appears to displace the VCA domain of N-WASP to stimulate branch formation, and then to remain associated with—and stabilize—the growing branch. Helgeson and Nolen suggest that these NPFs work together to create branches using an "obligatory displacement" model. According to this scheme, N-WASP (or another type I NPF), the Arp2/3 complex and two actin molecules are bound at the site of a future branch on an actin filament, poised for branch formation. However, before more actin molecules can be added, N-WASP must be released, either slowly on its own—as Smith et al. also report in findings published concurrently in *eLife*—or rapidly with the help of cortactin or other type II NPFs.

Although the rationale for N-WASP removal is not yet understood, type I NPFs are generally attached to the plasma membrane. When N-WASP releases the mother filament, the membrane should no longer be able to block the addition of actin molecules to a growing branch.

due to differential engagement of the two sites (*Padrick et al., 2008*). Oligomerization of WASP/Scar proteins is thought to tune NPF activity in vivo (*Padrick et al., 2008*; *Gohl et al., 2010*; *Footer et al., 2008*), so dissecting biochemical differences between monomeric and oligomeric NPFs is an important challenge.

Cortactin, the prototypical type II NPF, was initially discovered as a Src kinase substrate and actin binding protein (*Wu and Parsons, 1993*), and was later found to directly bind and activate Arp2/3 complex (*Weed et al., 2000*). Cortactin contains an N-terminal acidic region (NtA), which interacts with Arp2/3 complex, 6.5 actin filament binding repeat sequences, and a C-terminal SH3 domain (*Figure 1B*). Mutations in the NtA that block its interaction with Arp2/3 complex prevent assembly of actin in protrusive structures in transformed cells called invadopodia, and hinder actin-dependent vesicle trafficking required for lamellipodial protrusion and cellular motility (*Bryce et al., 2005*; *Ayala et al., 2008*). These observations demonstrate that cortactin plays an important role in regulating Arp2/3 complex in vivo, yet the precise mechanism of Arp2/3 complex activation is unclear. Because the actin filament-binding repeats are required for activation, it has been hypothesized that cortactin recruits Arp2/3 complex to filaments to stimulate nucleation (*Uruno et al., 2001*) (*Figure 1C*). However, whether filament recruitment can explain the acceleration of branching nucleation is unknown. In addition, cortactin on its own is a weak activator of Arp2/3 complex in vitro (*Weed et al., 2000*), making it uncertain how cortactin can contribute to branching nucleation in vivo.

Importantly, in the presence of WASP/Scar proteins, cortactin potently activates Arp2/3 complex (*Weaver et al., 2001*; *Uruno et al., 2001*), suggesting these NPFs synergize to assemble branched actin structures in vivo (see *Figure 1C* for schematic). Consistent with this hypothesis, WASP/Scar proteins and cortactin co-localize in many branched networks in vivo, including at the leading edge of motile cells, podosomes and invadopodia, and at sites of endocytosis (*Martinez-Quiles et al., 2004*;

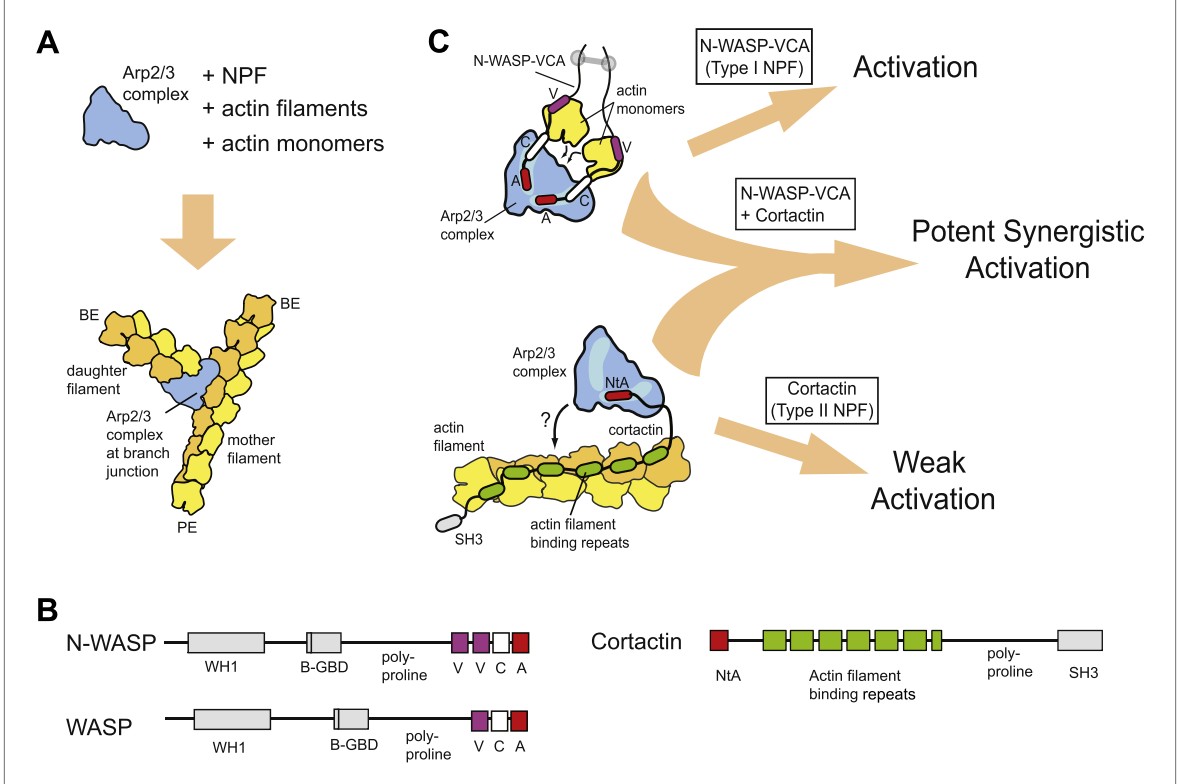

**Figure 1**. Schematic overview of branching nucleation and the proteins involved. (**A**) Overview of branching nucleation depicting the required reaction components (Arp2/3 complex, NPF, actin monomers and actin filaments) and the resultant Y-shaped branches. The barbed and pointed ends of the actin filaments are labeled BE and PE, respectively. (**B**) Domain organization of prototypical type I (WASP/N-WASP) and type II (Cortactin) NPFs. WH1, WASP homology 1; B-GBD, Basic region and GTPase binding domain; V, verprolin homology (also known as WH2, WASP homology 2); C, central; A, acidic; NtA, N-terminal acidic region; SH3, Src homology 3. (**C**) Schematic overview of activation of Arp2/3 complex by two classes of nucleation promoting factors. Gray barbell indicates a generic N-WASP dimerization mechanism. Small black arrows indicate either recruitment of actin monomers to Arp2/3 complex by VCA, or recruitment of Arp2/3 complex to actin filaments by cortactin. Light blue areas on Arp2/3 complex indicate the two proposed CA binding sites. The SH3 domain of cortactin is shown here but is omitted in other figures for clarity.

*DesMarais et al., 2009*; *Grassart et al., 2010*). Previous data showed that N-WASP and cortactin compete for binding to Arp3 (*Weaver et al., 2002*), and cortactin competes more strongly when Arp2/3 complex is bound to filaments (*Uruno et al., 2003*), but the precise mechanism of synergy is unknown. Previously proposed models include scenarios in which cortactin recruits Arp2/3 complex to the mother filament, where it cooperates with VCA to activate nucleation or induces release of VCA from branch-incorporated Arp2/3 complex to stabilize the nucleus (*Weaver et al., 2002*; *Uruno et al., 2003*). In another model, VCA becomes sequestered at branch junctions, so the concentration of VCA available to activate Arp2/3 complex limits the rate of branching nucleation (*Siton et al., 2011*). In this model, called the recycling model, cortactin binding displaces VCA from branch junctions, recycling it back into solution for more activation.

Here we dissect the mechanism of synergy between N-WASP and cortactin. Using single-molecule total internal reflection fluorescence (smTIRF) microscopy along with biochemical assays and mathematical modeling, we show that neither a filament recruitment nor a VCA recycling model can explain synergy. Our data instead support an obligatory displacement model, in which cortactin directly targets nascent branch junctions to accelerate the release of VCA. A key concept of the model is that VCA release is required for nucleation, either with or without cortactin, and the VCA release rate modulates the rate of nucleation. We dissect the biochemical requirements for synergy in cortactin and N-WASP to show that oligomerization of N-WASP VCA is required for significant synergy, but the actin filament binding repeats of cortactin are not. In addition, we provide evidence that slow release of N-WASP at nascent branch junctions limits nucleation rates, and that synergy is dependent on the

ability of cortactin to accelerate this step. Our data provide important mechanistic insights into the regulation of Arp2/3 complex by cortactin, and lay the foundation for a molecular understanding of how NPFs work together to regulate branching nucleation.

## Results

### Cortactin-mediated synergy follows a hyperbolic activity curve

Previous experiments showed that cortactin activates Arp2/3 complex weakly on its own, but potently synergizes with WASP/N-WASP (*Weed et al., 2000*; *Weaver et al., 2001*; *Uruno et al., 2001*). To quantify synergy, we added a range of concentrations of cortactin to a reaction with GST-N-WASP-VCA (GST-VCA) and Arp2/3 complex (*Figure 2A*). Cortactin dramatically increased the polymerization rate, and the concentration dependence of synergy followed a hyperbolic trend (*Figure 2B*). The concentration of cortactin required for half-maximal synergy was 72 nM and saturating cortactin increased the maximum polymerization rate 3.5–3.8-fold over GST-VCA alone. In contrast to a previous report, cortactin did not inhibit Arp2/3 complex at any concentration we tested, up to 20 µM in pyrene actin polymerization assays or 1 µM in TIRF microscopy branching assays (*Siton et al., 2011*) (*Figure 1B–D* and *Video 1*).

### Actin filament recruitment cannot account for synergistic activation

Several lines of evidence suggest cortactin might synergize with GST-VCA by recruiting Arp2/3 complex to actin filament sides. First, Arp2/3 complex must bind to the side of a pre-existing filament to

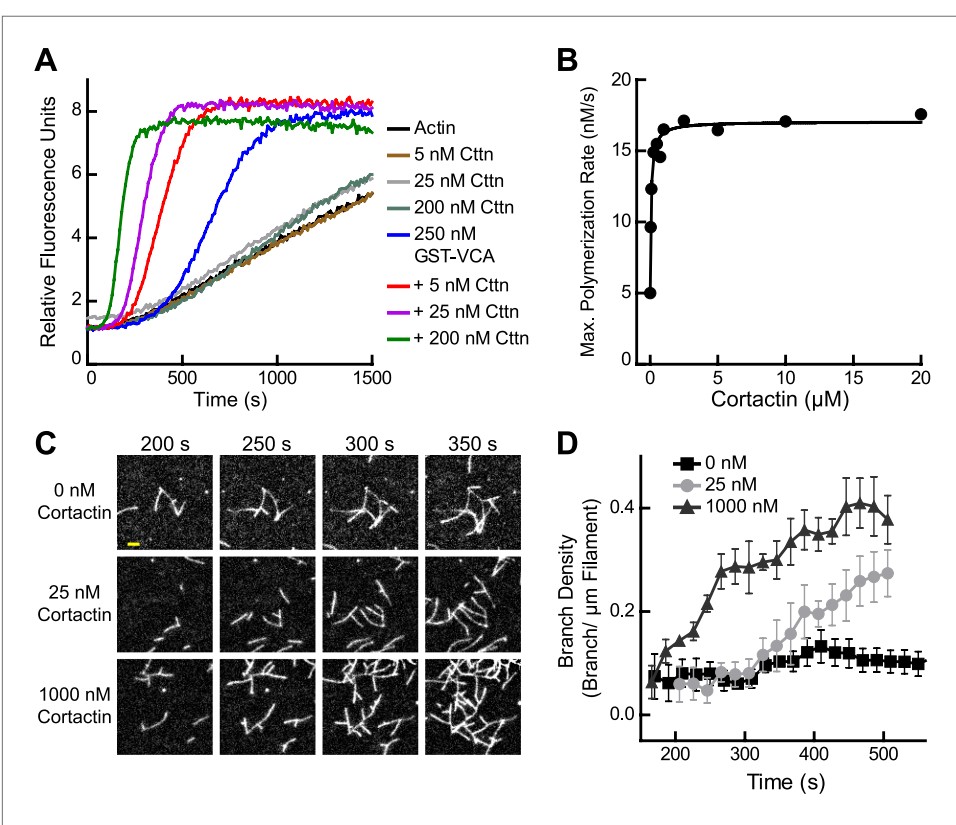

**Figure 2**. Cortactin synergizes with GST-N-WASP-VCA. (**A**) Time course of pyrene-actin polymerization showing synergistic activation of Arp2/3 complex by cortactin and GST-VCA. Reactions contain 2 µM 15% pyrene-actin, 20 nM Arp2/3 complex and cortactin (Cttn) and/or 250 nM GST-VCA as indicated. (**B**) Plot of maximum polymerization rate vs cortactin concentration for reactions conditions as in panel **A** with 150 nM GST-VCA. Data were fit as described in 'Materials and methods'. (**C**) TIRF microscopy images of reactions containing 1 µM 33% Oregon-Green actin, 10 nM Arp2/3 complex, 50 nM GST-VCA and indicated concentrations of cortactin. (**D**) Branch density time vs time for TIRF data from panel **C**. Error bars are the standard error of the mean for at least three regions of interest from an acquisition period. Scale bar: 2 µm.

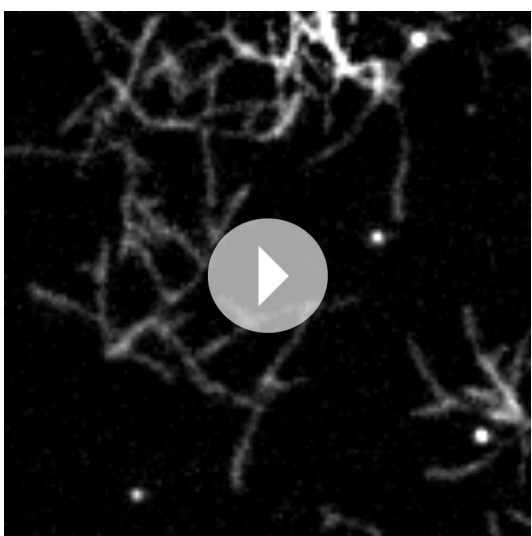

**Video 1**. Synergistic activation of Arp2/3 complex by GST-VCA and high concentrations of cortactin. Video corresponds to images in **Figure 2C**. Reaction contains 1 µM 33% Oregon-Green actin, 10 nM Arp2/3 complex, 50 nM GST-VCA and 1 µM cortactin. Single-wavelength (488 nm) images were acquired at a final magnification of 100× with an exposure time of 100 ms and a frame rate of 1 fps (frames per second).

be activated (**Achard et al., 2010**), and kinetic measurements of pyrene-labeled *S. pombe* Arp2/3 complex indicated this binding step is slow (**Beltzner and Pollard, 2008**). Second, deletion of the actin filament binding domain of cortactin abolished its weak intrinsic nucleation activity (**Uruno et al., 2001**). Finally, cortactin increases copelleting of Arp2/3 complex with actin filaments (**Cai et al., 2008**). To determine if filament recruitment can explain synergistic activation, we first constructed a mathematical model to describe branching nucleation in the presence of GST-VCA and Arp2/3 complex without cortactin (**Figure 3A**, **Figure 3—figure supplement 1** and 'Materials and methods'). Our goal was to determine how the rate constants for actin filament binding by Arp2/3 complex influence polymerization time courses, and if recruitment of Arp2/3 complex by cortactin could account for the increased rates we measured in our bulk polymerization assays containing cortactin.

We fit pyrene-actin polymerization time courses of reactions containing 50 nM Arp2/3 complex with increasing concentrations of GST-VCA to a mathematical model similar to that of Beltzner et al. (**Beltzner and Pollard, 2008**). The final activation step ($k_{nuc}$) was optimized by globally fitting the data from time courses at a range of concentrations of GST-VCA, while fixing all other kinetic parameters (**Figure 3A**; **Tables 1 and 2**). The off rate of Arp2/3 complex bound to a mother filament ($k_{fil\_off}$) was constrained based on the measured $K_D$, 0.9 µM (**Hetrick et al., 2013**). The model assumes that the final activation step ($k_{nuc}$) occurs after two actin monomers have been recruited by GST-VCA (**Padrick et al., 2011**). The model fit the experimental data well, showing a good visual fit to the time courses and a low residual sum of squares ($1.3 \times 10^{-11}$) (**Figure 3B**). Optimization of the mother filament on rate showed that $k_{fil\_on}$ approaches a minimum threshold ($2.7 \times 10^3$ M$^{-1}$ s$^{-1}$), beyond which, increases in the on rate do not improve the fit (**Figure 3C**). We fixed the on rate at $1.4 \times 10^6$ M$^{-1}$ s$^{-1}$ based on modeling of reactions containing cortactin (see below and 'Materials and methods'), and determined the optimized value for $k_{nuc}$ is 0.0038 s$^{-1}$. This value is 320-fold smaller than the calculated off rate of the nascent branch complex from filament sides, consistent with smTIRF studies that show Arp2/3 complex binds and is released from filaments many times before nucleating a branch (**Smith et al., 2013**).

We then used the optimized model to determine if recruitment of Arp2/3 complex to the sides of filaments by cortactin can explain synergy. We mimicked the effect of recruitment by simulating an increase in the concentration of actin filaments to saturate Arp2/3 complex side binding. If synergy occurs purely through recruitment, the magnitude of polymerization rate increases caused by adding side-binding sites will be similar to cortactin-induced rate increases. Using $k_{fil\_on}$ determined from the experimental data with cortactin ($1.4 \times 10^6$ M$^{-1}$ s$^{-1}$, see below, blue line in **Figure 3C**), we found that increasing side binding sites increased the polymerization rate, but could not account for the dramatic rate increases we observed in experiments containing cortactin (**Figure 3D**). This suggests that filament recruitment cannot account for cortactin-mediated synergy. Because of the uncertainty in $k_{fil\_on}$, we repeated the simulations at additional $k_{fil\_on}$ values. First, we used a $k_{fil\_on}$ value ($6.9 \times 10^5$ M$^{-1}$ s$^{-1}$, purple line in **Figure 3C**) calculated from the experimentally determined filament off rate of budding yeast Arp2/3 complex (0.625 s$^{-1}$) and the affinity of the bovine complex for filaments (**Smith et al., 2013**; **Hetrick et al., 2013**). This simulation also failed to account for experimentally observed synergy. However, in a simulation at the minimum threshold on rate ($2.7 \times 10^3$ M$^{-1}$ s$^{-1}$, green line in **Figure 3C**), filament recruitment could fully account for synergy. Therefore, while our best estimates of $k_{fil\_on}$ suggest that actin filament recruitment cannot explain synergy, $k_{fil\_on}$ is not determined well enough to completely eliminate the possibility that synergy occurs through recruitment.

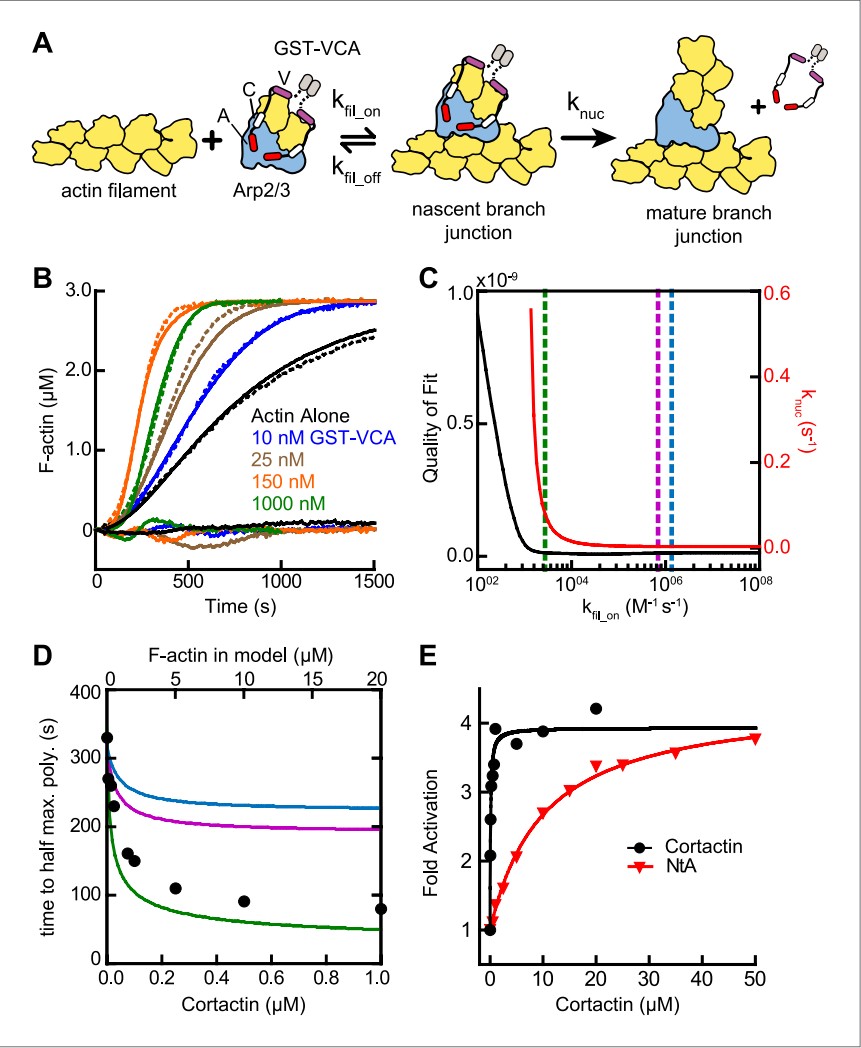

**Figure 3**. Actin filament recruitment cannot explain cortactin-mediated synergy. (**A**) Cartoon pathway of steps optimized in the kinetic model of branching nucleation. (**B**) Representative pyrene-actin polymerization time courses of Arp2/3 complex activated by GST-VCA (dashed lines) with simulated fits (solid lines). Residuals are shown below as solid lines. Reactions contained 3 µM 15% pyrene-actin, 50 nM Arp2/3 complex and indicated concentrations of GST-VCA. (**C**) Plot showing the relationship between the quality of fit (black line) and the optimized value of $k_{nuc}$ (red line) for simulations at a range of fixed values of $k_{fil\_on}$. Dashed purple and blue lines show $k_{fil\_on}$ values supported by our analysis in **Figure 7** (blue line) or by empirically measured $k_{off}$ and $K_D$ values (purple line) (**Hetrick et al., 2013**; **Smith et al., 2013**). The dashed green line indicates the minimum value of $k_{fil\_on}$ that fits the data with a quality of fit better than $1.3 \times 10^{-11}$. Quality of fit was calculated by a mean-weighted residual sum of squares. (**D**) Simulations showing the effect of increased actin filament side binding sites on the half time to reach equilibrium. Simulations were run using the three different $k_{fil\_on}$ values indicated in panel **C**. Empirical data from actin polymerization time courses with 3 µM 15% pyrene-actin, 20 nM Arp2/3 complex, 100 nM GST-VCA and the indicated concentrations of cortactin are shown as black circles (bottom axis). Initial concentrations of modeled actin filaments in the simulation are indicated on the top axis. (**E**) Plot of the fold activation over GST-VCA alone for a range of concentrations of full-length cortactin or NtA. Reactions contain 2 µM 15% pyrene-actin, 20 nM Arp2/3 complex, 250 nM GST-VCA and indicated concentrations of cortactin or NtA. Fold activation is calculated as the maximum polymerization rate for each reaction divided by the maximum polymerization rate for the reaction without cortactin. Data were fit (solid lines) as described in 'Materials and methods'.

The following figure supplements are available for figure 3:

**Figure supplement 1**. Mathematical modeling of actin polymerization in the presence or absence of GST-VCA, Arp2/3 complex and cortactin.

**Table 1.** Mathematical modeling parameters

| Reaction # | Description | $k_{on}$ (M⁻¹s⁻¹) | $k_{off}$ (s⁻¹) | $K_D$ (μM) | Reference |
|---|---|---|---|---|---|
| 1 | Actin dimerization | $1.98 \times 10^7$ | $5.26 \times 10^7$ | $2.6 \times 10^6$ | Mullins 1998, This study |
| 2 | Actin trimerization | $1.16 \times 10^7$ | $4.07 \times 10^5$ | $3.5 \times 10^4$ | Mullins 1998, This study |
| 3 | Spontaneous nucleation | $1110 - 1220$*,† | | | This study |
| 4 | Barbed end elongation | $1.16 \times 10^7$ | 1.4 | | Pollard 1986 |
| 5 | Barbed end elongation, actin monomer bound to GST-VCA | $1.16 \times 10^7$ | 1.4 | | Pollard 1986, Higgs 1999 |
| 6 | Barbed end elongation, two actin monomers bound to GST-VCA | $1.16 \times 10^7$ | 1.4 | | Pollard 1986, Higgs 1999 |
| 7 | Actin monomer binds GST-VCA | $5 \times 10^6$ | 3 | 0.6 | Marchand 2001, Beltzner 2008 |
| 8 | Actin monomer binds GST-VCA with bound actin | $5 \times 10^6$ | 3 | 0.6 | Marchand 2001, Beltzner 2008 |
| 9 | Actin monomer binds GST-VCA:actin₂ | $4.2 \times 10^4$ | 74.4 | $1.8 \times 10^3$ | This study |
| 10 | Actin monomer binds GST-VCA:actin₃ | $1.5 \times 10^7$ | 1.04 | 0.069 | This study |
| 11 | Actin monomer binds GST-VCA:actin₄ | $2 \times 10^7$ | 0.062 | 0.003 | This study |
| 12 | GST-VCA nucleation | $6.1 \times 10^{-8}$* | | | This study |
| 13 | Arp2/3 binds actin filament | $1.37 \times 10^6$ | 1.23 | 0.9 | Hetrick 2013 |
| 14 | GST-VCA binds Arp2/3 | $0.8 \times 10^6$ | 0.072 | 0.009 | Padrick 2008 |
| 15 | GST-VCA:actin binds Arp2/3 | $0.8 \times 10^6$ | 0.014 | 0.018 | Padrick 2008, Beltzner 2008, Kelly 2006 |
| 16 | GST-VCA:actin₂ binds Arp2/3 | $0.8 \times 10^6$ | 0.029 | 0.028 | Padrick 2008, Beltzner 2008, Kelly 2006 |
| 17 | Actin monomer binds GST-VCA:Arp2/3 | $2.5 \times 10^6$ | 3 | 1.2 | Marchand 2001, Beltzner 2008 |
| 18 | Actin monomer binds GST-VCA:actin:Arp2/3 | $2.5 \times 10^6$ | 3 | 1.2 | Marchand 2001, Beltzner 2008 |
| 19 | GST-VCA:actin₂:Arp2/3 binds actin filament ($k_{fil\_on}$) | $1.37 \times 10^6$ | 1.23 | 0.9 | Hetrick 2013 |
| 20 | GST-VCA binds Arp2/3:F-actin | $0.8 \times 10^6$ | 0.072 | 0.009 | Padrick 2008 |
| 21 | GST-VCA:Arp2/3 binds F-actin | $1.37 \times 10^6$ | 1.23 | 0.9 | Hetrick 2013 |
| 22 | GST-VCA:Arp2/3:actin binds F-actin | $1.37 \times 10^6$ | 1.23 | 0.9 | Hetrick 2013 |
| 23 | GST-VCA:Arp2/3:F-actin binds actin monomer | $2.5 \times 10^6$ | 3 | 1.2 | Marchand 2001, Beltzner 2008 |
| 24 | GST-VCA:Arp2/3:actin:F-actin binds actin monomer | $2.5 \times 10^6$ | 3 | 1.2 | Marchand 2001, Beltzner 2008 |
| 25 | Arp2/3 complex nucleation ($k_{nuc}$) | $0.004 - 0.006$* | | | This study |
| 26 | Cortactin binds actin filament | $1.21 \times 10^4$ | 0.063 | 5.21 | This study |
| 27 | Cortactin binds nascent branch junction | $2.0 \times 10^6$ | 0.034 | 0.017 | This study |
| 28 | Synergy displacement activation of Arp2/3 complex ($k_{dis}$) | 0.036* | | | This study |
| 29 | Synergy recycling, cortactin dissociates sequestered GST-VCA | $2.0 \times 10^6$ | 0.034 | 0.017 | This study |

*Units are s⁻¹.

†This value was adjusted for each full set of reactions.

Underlined values were allowed to float during some optimizations, see **Table 2**.

**Table 2.** Mathematical modeling reaction sets

| Reaction set | Reactions | Initial concentrations | Variable concentrations (µM) | Floated parameters | Quality of fit |
|---|---|---|---|---|---|
| 1 | 1–4 | – | 2.0, 3.0, 4.0, 5.0, 6.0 actin monomers | $k_1$, $k_{-1}$, $k_2$, $k_{-2}$, $k_3$ | $1.75 \times 10^{-11}$ |
| 2 | 1–12 | 3 µM actin monomers | 0, 0.02, 0.04, 0.08, 0.1, 0.2, 0.6, 0.8, 1.0 GST-VCA | $k_9$, $k_{-9}$, $k_{10}$, $k_{-10}$, $k_{11}$, $k_{-11}$, $k_{12}$ | $2.32 \times 10^{-11}$ |
| 3 | 1–25 | 3 µM actin monomers, 50 nM Arp2/3 complex | 0, 0.01, 0.025, 0.050, 0.1, 0.15, 0.25, 0.5, 1.0 GST-VCA | $k_{25}$ ($k_{nuc}$) | $1.28 \times 10^{-11}$ |
| 4a | 1–28 | 3 µM actin monomers, 20 nM Arp2/3 complex, 100 nM GST-VCA | 0, 0.005, 0.025, 0.075, 0.1, 0.25, 1.0 cortactin | $k_{fil\_on}$($k_{13}$= $k_{19}$=$k_{21}$=$k_{22}$)*†, $k_{25}$†, $k_{28}$ ($k_{dis}$) | $2.87 \times 10^{-11}$ |
| 4b | 1–25, 29 | 3 µM actin monomers, 20 nM Arp2/3 complex, 100 nM GST-VCA | 0, 0.005, 0.025, 0.075, 0.1, 0.25, 1.0 cortactin | $k_{29}$, $k_{-29}$ | $4.26 \times 10^{-10}$ |

*$k_{fil\_on}$ is a single global variable used for the indicated reaction rates.
†Only optimized for the 0 µM cortactin reaction.

Therefore, we next asked if the actin filament binding repeats of cortactin are required for synergy. We tested a range of concentrations of full-length cortactin and a construct containing only the NtA (residues 1–84) for their ability to synergize with GST-VCA. We found NtA was synergistic with GST-VCA, demonstrating that actin filament binding is not required for synergy (*Figure 3E*). The concentration dependence of synergy for both constructs followed a hyperbolic trend, and the concentration of NtA required for half-maximal synergy was 11 µM, 110-fold higher than for full-length cortactin. However, at saturation, NtA was as potent as full-length cortactin, increasing the maximum polymerization rate 3.8-fold over GST-VCA-mediated activation of Arp2/3 complex. These data demonstrate that the NtA is sufficient for synergy, but that the actin filament binding repeats allow cortactin to synergize at lower concentrations.

### The oligomerization state of the type I NPF is an important determinant of synergy

To probe further the mechanism of synergy, we next asked if biochemical features of the type I NPF influence synergy. Dimerization of VCA is known to increase its binding affinity for Arp2/3 complex, and some evidence suggests WASP/Scar proteins function as oligomers in vivo (*Padrick et al., 2008*; *Gohl et al., 2010*; *Footer et al., 2008*). Therefore, we compared cortactin-mediated synergy with monomeric vs GST-tagged N-WASP-VCA in a pyrene actin polymerization assay. In previous experiments, GST-VCA behaved similarly to WASP dimerized through physiological SH3-polyproline interactions, so artificial dimerization by GST can mimic in vivo dimerization mechanisms (*Padrick et al., 2008*). Saturating cortactin enhanced the maximum polymerization rate of a reaction containing GST-VCA 3.7-fold over the rate without cortactin, whereas cortactin weakly influenced a reaction containing monomeric VCA, accelerating the reaction only ~1.5-fold over VCA alone (*Figure 4A,B*, *Figure 4—figure supplement 1*). Increasing the concentration of monomeric VCA did not increase synergy, suggesting the failure to observe potent synergy is not due to under-saturation of two CA binding sites on the complex (*Figure 4B,C*, *Figure 4—figure supplement 1*). VCA dimerized with a leucine zipper (LZ-VCA) behaved identically to GST-VCA, demonstrating that the difference in synergy is due to the oligomerization state of the VCA rather than an artifact caused by GST (*Figure 4A*).

The dimerization state of N-WASP could control synergy by influencing the number of actin monomers recruited to the complex. For instance, if NtA and VCA simultaneously interact with Arp2/3 complex during activation, GST-VCA may be able to recruit two actin monomers while engaging only one NPF binding site, while VCA can only recruit one actin monomer (*Figure 4D*). Unlike WASP and Scar, native N-WASP contains tandem V regions and may be able to recruit two actin monomers to Arp2/3 complex (*Rebowski et al., 2010*). Therefore, we made a monomeric N-WASP construct with both V regions (VVCA) and tested its ability to synergize with cortactin. We found that VVCA is not

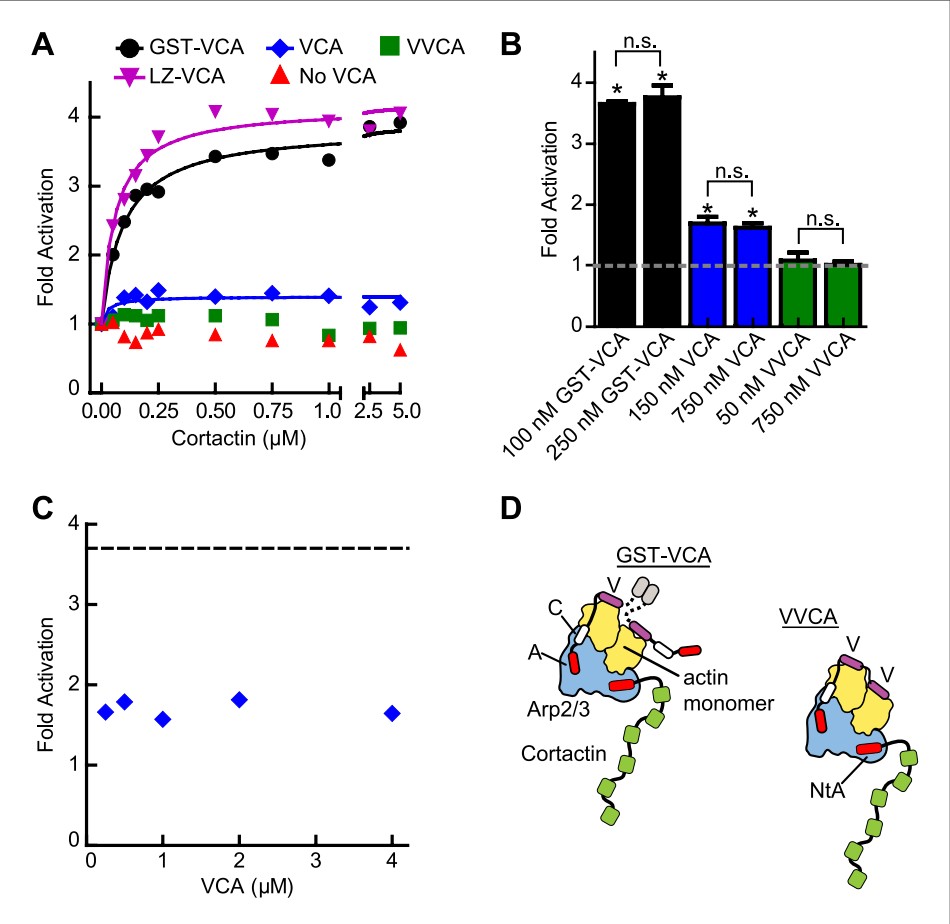

**Figure 4**. The oligomerization state of VCA is an important determinant of synergy. (**A**) Plot of the fold activation vs cortactin concentration for reactions containing 2 µM 15% pyrene-actin, 20 nM Arp2/3 complex and 250 nM GST-VCA (black), 750 nM VCA (blue), 750 nM VVCA (green), 250 nM Leucine-zipper VCA (LZ-VCA, magenta) or no N-WASP (red). Monomer concentrations are listed. Fold activation is calculated as described in *Figure 3E*. (**B**) Plot of the average fold activation for reactions containing 1 µM cortactin and the indicated concentration of GST-VCA, VCA or VVCA. Dashed line indicates no synergy. p-values were calculated by two-tailed Student's *t*-test. Error bars are S.E.M. n.s. = not significant, p>0.05. Asterisks indicate average fold activation values are significantly different (p<0.05) than a fold activation value of 1 (no synergy). (**C**) Fold activation vs concentration of monomeric VCA for pyrene actin polymerization assays containing 0 to 4 µM N-WASP-VCA, 500 nM cortactin and Arp2/3 complex and actin as in panel **A**. Dashed line shows average fold activation for a reaction containing 250 nM GST-VCA and saturating cortactin. (**D**) Cartoon showing hypothetical branching intermediates with the potential to recruit two actin monomers to Arp2/3 complex with bound cortactin.

The following figure supplements are available for figure 4:

**Figure supplement 1**. The level of synergy is the same at saturating and subsaturating concentrations of the type I NPF.

synergistic with cortactin at any concentration (*Figure 4A,B*, *Figure 4—figure supplement 1*). These data suggest that the ability to recruit two actin monomers to the complex using one NPF binding site is not sufficient for synergy.

## Dissociation of N-WASP from branch junctions may limit the rate of branching nucleation

Dimerization increases the affinity of N-WASP for Arp2/3 complex 180-fold, allowing it to saturate both NPF binding sites on the complex at lower concentrations than monomeric VCA (*Padrick et al., 2008*). We next explored the importance of the affinity of the type I NPF for Arp2/3 complex in

synergy. Based on a number of observations, we hypothesized that type I NPF release is required for nucleation, and that tight binding of the type I NPF can slow the final activation step. First, we observed that at saturation, VCA is a better activator of Arp2/3 complex than GST-VCA (*Figure 5A*). Second, recent single-molecule imaging experiments show that WASP dissociates from the branch junctions before elongation of the daughter filament (*Martin et al., 2006*) (*Smith et al., 2013*, in press at eLife). Finally, crystal structures show that the V region may block the barbed end of actin monomers, preventing interactions with incoming actin monomers required for elongation (*Chereau et al., 2005*). Therefore, we hypothesized that cortactin synergizes with GST-VCA by displacing it from the nascent branch complex, increasing the rate of the final activation step by accelerating the obligatory release of GST-VCA (*Figure 5B*).

To test this model, we prepared a VCA construct (GST-VCA(R442A)) with a point mutation in V that decreases its affinity for actin monomers 20-fold (*Co et al., 2007*) (*Figure 5B,C*). We reasoned that this mutation would decrease the affinity of GST-VCA for the nascent branch junction, but still allow V to recruit actin monomers to the Arp2/3 complex (*Co et al., 2007*). We predicted that in the absence of

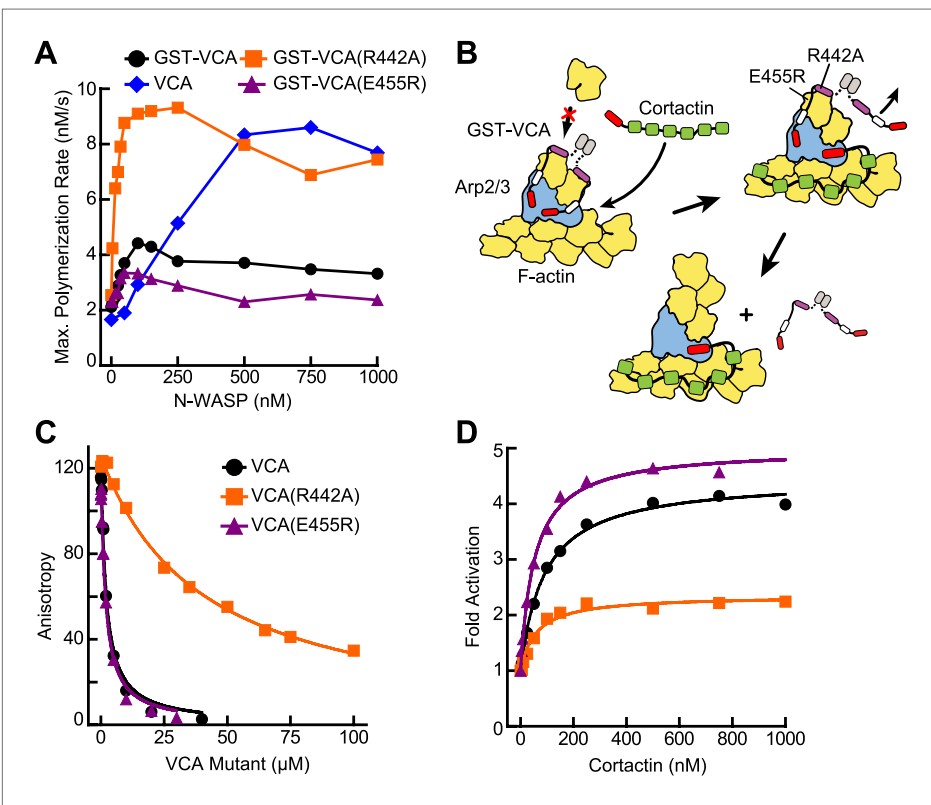

**Figure 5**. VCA affinity for the nascent branch junction is an important determinant of synergy. (**A**) Maximum polymerization rates verses (monomer) concentration of N-WASP constructs for reactions containing 20 nM Arp2/3 complex and 2 µM 15% pyrene-actin. (**B**) Cartoon depicting obligatory displacement model of cortactin-mediated synergy. Approximate location of residues (based on 2A41.pdb and 2VCP.pdb) (*Chereau et al., 2005*; *Gaucher et al., 2012*) mutated in the V region of GST-VCA are indicated in one V region. (**C**) Fluorescence anisotropy binding measurements showing competition between wild type rhodamine-VCA and unlabeled VCA constructs for actin monomers. $K_{D, WT}$ = 0.56 ± 0.03 µM (black) $K_{D, R442A}$ = 11.2 ± 1.3 µM, (orange) $K_{D, E455R}$ = 0.63 ± 0.06 µM (purple). (**D**) Plot of fold activation vs cortactin concentration for reactions containing 20 nM Arp2/3 complex, 2 µM 15% pyrene-actin and 250 nM of a GST-VCA construct, colors as in panel **A**. Fold activation is calculated as described in *Figure 3E*.

The following figure supplements are available for figure 5:

**Figure supplement 1**. The E455R mutation in N-WASP is predicted to provide additional favorable electrostatic interactions with actin monomers.

cortactin, saturating concentrations of this mutant would have a higher maximal polymerization rate, because it would release from the nascent branch junction faster than wild-type GST-VCA. Consistent with this prediction, the maximal polymerization rate at saturation was increased 2.5-fold in the R442A mutant compared to wild type (*Figure 5A*). The obligatory displacement model predicts that cortactin will be less synergistic with the GST-VCA(R442A) mutant because of its higher intrinsic off rate from the nascent branch complex. Our data are also consistent with this prediction. In the presence of the R442A mutant, cortactin showed approximately twofold less synergy compared to the wild-type GST-VCA (*Figure 5D*). As an additional test of the model, we made a V region mutation, E455R, which we predicted would bind more tightly to actin monomers and the nascent branch junction. Based on mutational data and a crystal structure of the V region of WIP bound to actin, an E455R mutation was predicted to interact favorably with Glu93 on the surface of actin (*Didry et al., 2011*) (*Figure 5—figure supplement 1*). Interestingly, this mutation had no influence on the affinity of N-WASP-VCA for actin monomers (*Figure 5C*). However, it decreased the maximal polymerization rate at saturation (*Figure 5A*) and increased synergy with cortactin (*Figure 5D*). This suggests that the E455R/E93 interaction may occur only in the context of the nascent branch junction. Together, these data support an obligatory displacement model for synergistic activation of the complex.

## Cortactin binds statically to actin filaments

Our data suggest that cortactin may displace GST-VCA from branch junctions to activate the complex synergistically. This model requires that cortactin bind to nascent branch junctions and avoid being nonproductively sequestered along the sides of filaments. The actin filament-binding region of cortactin is composed of 6.5 tandem 37-amino acid repeats, which are unstructured (*Shvetsov et al., 2009*). The multivalent architecture of its actin binding domain led us to hypothesize that cortactin may find branch junctions by diffusing along actin filaments through multiple weak and dynamic interactions, similar to the actin binding protein VASP (*Hansen and Mullins, 2010*). To test this, we labeled cortactin (residues 1–336) with Alexa-568 and actin monomers with Oregon-Green 488 and visualized their interactions using TIRF microscopy. Single-molecules of cortactin bound to and dissociated from actin filaments during the time courses of the videos (*Figure 6A*, *Figure 6—figure supplement 1*; *Video 2*). The cortactin molecules bound statically, eliminating the possibility that cortactin targets nascent branch junctions by diffusing along filaments (*Figure 6B*). We measured the lifetimes of bound cortactin molecules and fit the cumulative lifetime data to a single exponential decay equation and determined an off rate ($k_{off}$) of 0.050 s$^{-1}$ (*Figure 6C*).

## Cortactin specifically targets preformed branch junctions

Because cortactin does not diffuse along filaments to find branch junctions, we hypothesized that cortactin may target junctions simply by preferentially binding junctions over filament sides. To test this, we visualized single molecules of cortactin with preformed branch junctions. Cortactin added to a TIRF chamber with preformed branches bound to both filament sides and branch junctions (*Figure 7A*, *Video 3*). The off rate of cortactin for branch junctions was 0.034 s$^{-1}$, almost twofold slower than the off rate for preformed filament sides measured from the same reaction (0.063 s$^{-1}$) (*Figure 7B*). Interestingly, 30% of the 270 tracked cortactin molecules were bound to branch junctions even though branches made up only ~0.13% (68 branch junctions vs ~53,000 filament side binding sites) of total cortactin binding sites (*Figure 7C,D*). This suggests that cortactin binds with a significantly higher affinity to branch junctions than filament sides. A twofold change in the off rate is unlikely to account for this difference. To determine if cortactin binds to branch junctions with a higher on rate than filament sides, we first measured the affinity of cortactin for branch junctions and filament sides using the single-molecule data. We counted the total number of bound cortactin molecules, branch junctions and side binding sites in each frame and calculated the average fraction of cortactin bound over hundreds of frames. Using the average fraction bound, we determined that the affinity of cortactin for branch junctions is ~300-fold greater than for filament sides: 17 nM vs 5.2 μM, respectively (*Figure 7E*). Using these equilibrium constants and our previously measured off rates, the calculated on rates of cortactin for branch junctions and filament sides are 2.0 × 10$^6$ M$^{-1}$ s$^{-1}$ and 1.2 × 10$^4$ M$^{-1}$ s$^{-1}$, respectively. Our data show that cortactin targets branch junctions by binding over two orders of magnitude more tightly to junctions than filament sides. High affinity binding to branch junctions is accomplished through a ~160-fold increase in the on rate and further amplified by a ~twofold decrease in the off rate. These data explain how cortactin can specifically target nascent branch junctions to displace N-WASP.

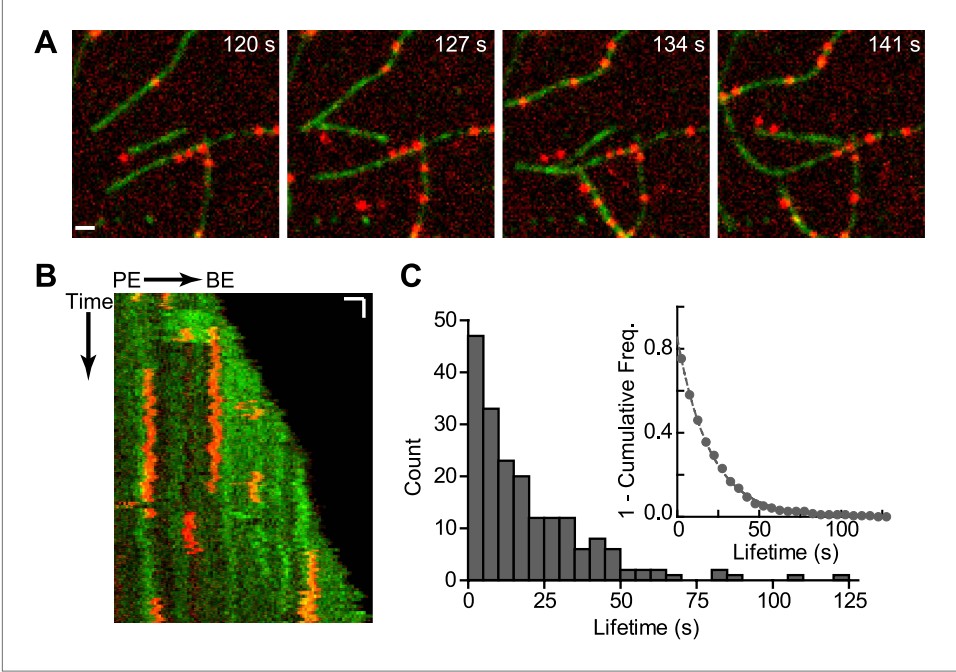

**Figure 6**. Cortactin binds statically to actin filaments. (**A**) smTIRF microscopy images of single cortactin molecules (red) bound to polymerizing actin filaments (green). TIRF reactions contained 1 µM 33% Oregon-Green actin and 2 nM Alexa568-cortactin (residues 1–336). Scale bar: 1 µm. (**B**) Kymograph showing cortactin molecules bound statically to a polymerizing filament. The barbed end (BE) and pointed end are indicated. Vertical scale bar: 10 s, horizontal scale bar: 1 µm. (**C**) Histogram showing binned lifetimes of single molecules of cortactin on actin filament sides. Counts were transformed into 1-cumulative frequency plot (inset) and fit with a single-exponential decay equation to determine the off rate, 0.050 s$^{-1}$ n = 191.

The following figure supplements are available for figure 6:

**Figure supplement 1**. Validation of single molecule data.

## Cortactin remains at the branch junction during daughter filament elongation

The obligatory displacement model predicts that cortactin can target nascent branch junctions, displace N-WASP, and remain bound to the branch junction without blocking elongation of the new (daughter) filament (***Figure 5B***). To test this prediction, we visualized single-molecules of cortactin in a reaction during active branching nucleation. We observed multiple instances (n = 66) in which cortactin bound to the side of a filament where a daughter filament was later nucleated (***Figure 8A,B***, ***Videos 4 and 5***). In each instance, cortactin remained bound to the branch junction after nucleation, consistent with the predictions of the obligatory displacement model. The average lifetime at the junction after nucleation was 62 s, 2.1-fold longer than the average lifetime of cortactin binding to a preformed branch junction (***Figure 8C***). The average lifetime before nucleation was 6.5 s. While the accuracy of pre- and post-nucleation lifetime measurements is limited by our ability to resolve newly formed daughter filaments, the data clearly indicate that cortactin stays bound to the junctions long after nucleation. During active branching, cortactin bound existing branch junctions with an average lifetime of 29.5 s, the same lifetime found using preformed filament reactions (29.1 s). Interestingly, the lifetime of cortactin molecules on filament sides was 2.0–2.5-fold lower (8.0 s) than with preformed filaments or polymerizing filaments in the absence of Arp2/3 complex. We cannot currently explain this result, but speculate that it may be due to conformational changes in the filament caused by Arp2/3 complex binding.

These data show that cortactin remains at the branch junction during and after nucleation, but do not allow us to eliminate the possibility that NtA disengages from Arp2/3 complex during elongation while the actin filament binding repeats hold cortactin at the junction. To test this, we labeled an NtA

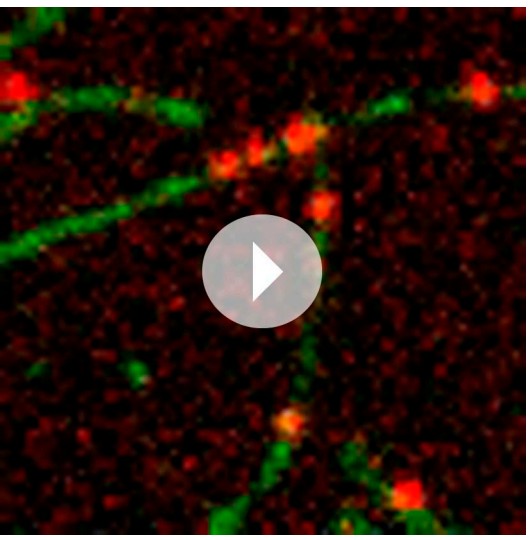

**Video 2**. Single molecules of cortactin interacting with polymerizing actin filaments. Reaction contains 1 μM 33% Oregon-Green actin (green) and 2 nM Alexa568-cortactin (red). Images from both channels were acquired using a 50 ms exposure at a ratio of 5:1 and a frame rate of 3.7 frames per second (fps) and 1.6 fps for the 488- and 561-channels, respectively.

fragment (residues 1–48) with Alexa-568 and visualized it during branching reactions. Single-molecules of NtA were observed bound to existing branch junctions and to nascent branches from which new daughter filaments nucleated (**Figure 8C** and **Video 6**). These data indicate that NtA can directly engage Arp2/3 complex after and during branch nucleation and that it does not block daughter filament elongation. This observation is consistent with pyrene actin polymerization assays showing NtA does not inhibit nucleation even at high micromolar concentrations (**Figure 3E**) (**Weaver et al., 2002**).

## An obligatory displacement model can account for the influence of cortactin in pyrene actin polymerization assays

Given the biochemical evidence in support of the displacement mechanism, we next built a mathematical model to determine if obligatory displacement could account for the synergy we observed in our bulk polymerization assays. This model was similar to the recruitment model described above, but included additional reactions to account for cortactin interactions with filament sides and at nascent branch junctions (**Figure 9A**, **Figure 3—figure supplement 1** and 'Materials and methods'). We used the kinetic rate constants determined from our single-molecule experiments to describe these reactions, and assumed that cortactin binds to the nascent branch junction with the same rate constants as mature branch junctions. Following cortactin binding to the nascent branch junction, we added a cortactin-mediated displacement activation step ($k_{dis}$) analogous to the GST-VCA-dependent activation step ($k_{nuc}$). Full time courses of pyrene-actin polymerization at a range of cortactin concentrations were fit with the new model, allowing only $k_{dis}$ and $k_{fil\_on}$ to float while $k_{fil\_off}$ was constrained by the previously measured $K_D$ value (**Figure 9B**). This model fit the experimental data well (**Figure 9B,C**). A plot of the half time to reach equilibrium ($t_{1/2}$) vs cortactin concentration indicated that $t_{1/2}$ decreased identically in both the model and the experimental data, and both showed a $t_{1/2}$ of about 80 s at saturating cortactin (**Figure 9D**). As in the simulations of reactions without cortactin, $k_{fil\_on}$ had a minimal threshold value, but in these simulations the minimum value was ~500-fold greater, $1.4 \times 10^6$ M$^{-1}$ s$^{-1}$ (**Figure 9C**). Above the threshold, the optimized value for displacement activation ($k_{dis}$) was 0.036 s$^{-1}$, 10-fold higher than the activation step in the absence of cortactin, $k_{nuc}$ (**Figure 9C**). The simplifications and assumptions made in the construction of the model limit the precision of the optimized values in representing the true microscopic rate constants. Nevertheless, the kinetic pathway of the obligatory displacement model fits the experimental data well, indicating that this model can explain synergy.

A previously proposed model for cortactin-mediated synergy hypothesized that cortactin displaces GST-VCA from the Arp2/3 complex during elongation of the daughter filament instead of before or during nucleation (**Siton et al., 2011**). In this model, called the recycling model, cortactin is hypothesized to prevent GST-VCA from being sequestered at branches, which could slow polymerization by limiting the concentration of GST-VCA in solution available to activate Arp2/3 complex. A key prediction of this model is that the concentration of free GST-VCA limits branching nucleation rates. To test this mechanism, we simulated recycling by forcing GST-VCA to remain bound after nucleation and introducing a single binding reaction whereby cortactin returns the branch junction-bound GST-VCA to the pool of non-sequestered GST-VCA (**Figure 3—figure supplement 1**). We attempted to use this model to fit time courses of pyrene actin polymerization containing Arp2/3 complex, GST-VCA and a range of concentrations of cortactin. The recycling displacement model could not fit the data because increasing

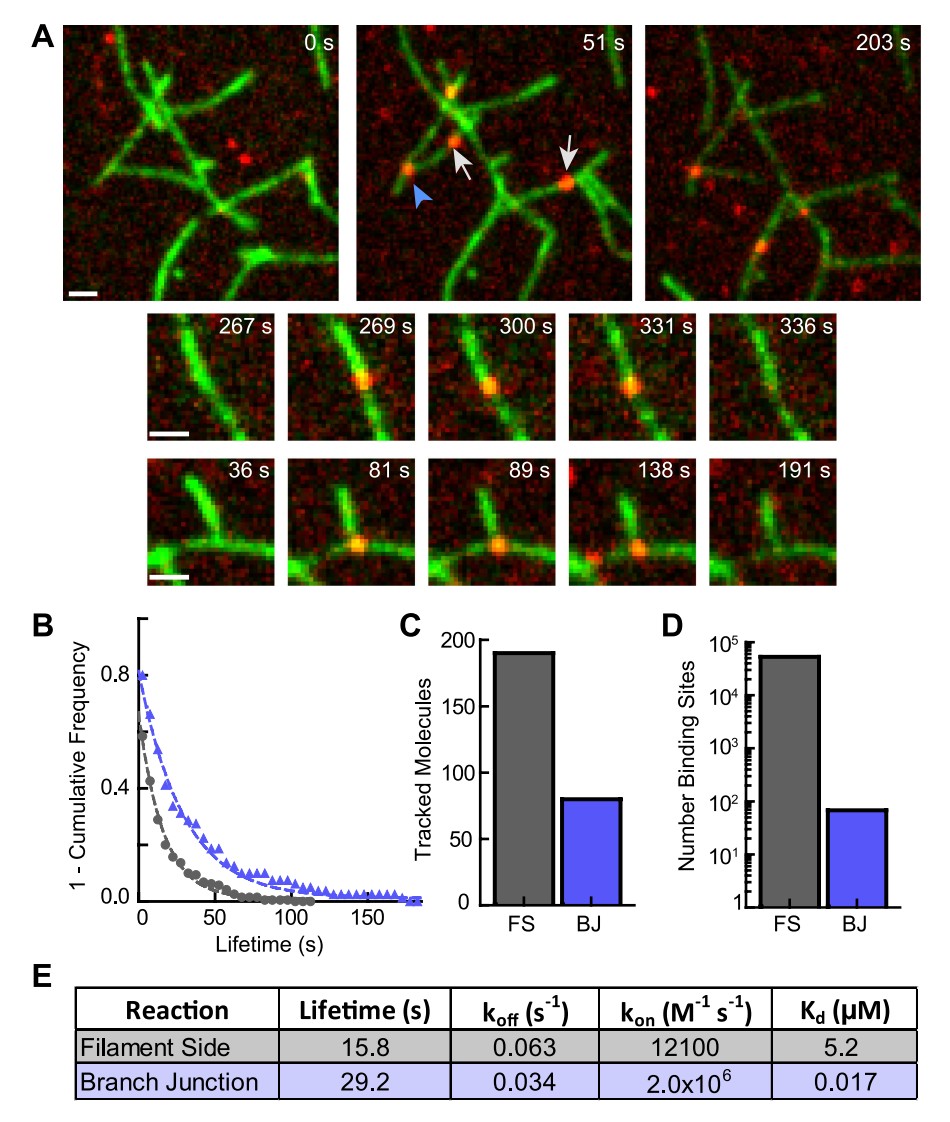

**Figure 7**. Cortactin directly targets branch junctions with a fast on rate. (**A**) smTIRF microscopy images showing interaction of cortactin with preformed branched networks. Reactions were initiated using 1 µM 33% Oregon-Green actin, 5 nM Arp2/3 complex and 30 nM VCA and allowed to proceed for ~6 min before flushing a solution containing 1.5 nM Alexa568-cortactin and 0.1 µM actin monomers into the reaction chamber. Single cortactin molecules (red) bound actin filament sides (gray arrows) and branch junctions (blue arrowhead). Large images show a single region of interest with both side and branch binding events. Time after cortactin addition is indicated. Smaller images show examples of complete filament side and branch junction binding events. Scale bars: 1 µm. (**B**) Frequency plot of tracked cortactin lifetimes for molecules bound to filament sides (gray) or branch junctions (blue) and fit with a single exponential decay function. (**C**) Plot of the total number of tracked cortactin molecules on filament sides (FS) or branch junctions (BJ). (**D**) Plot of the average number of cortactin filament side or branch binding sites across all analyzed frames. (**E**) Summary of kinetic and thermodynamic binding constants for each class of binding event.

concentrations of cortactin had no influence on modeled polymerization rates. To determine why, we plotted the concentration of sequestered GST-VCA during the modeled time courses at various concentrations of cortactin. In the absence of cortactin, only 3.8% of the total GST-VCA was sequestered at the end of the reaction (**Figure 9E**). Addition of cortactin reduced the concentration of sequestered GST-VCA, but did not influence the polymerization rate because the free GST-VCA concentration did not limit the rate of the reaction. Therefore, the recycling model cannot account for the synergistic

activation of Arp2/3 complex we observe in our assays.

## Discussion

Here we dissect the mechanism of synergistic activation of Arp2/3 complex by cortactin and N-WASP. Our data support an obligatory displacement model in which cortactin specifically targets nascent branch junctions to displace GST-VCA, thereby accelerating nucleation (*Figure 9*). In this model, GST-VCA, two actin monomers and Arp2/3 complex assemble on the side of a filament, creating a nascent branch junction. Cortactin targets the junction, making a multivalent interaction with Arp2/3 complex and the adjacent mother filament. At the nascent branch junction, NtA can compete with CA at the Arp3 binding site on the complex, as previously reported (*Weaver et al., 2002*; *Padrick et al., 2008*). Release of one CA speeds release of the GST-VCA, allowing nucleation and elongation to proceed. Importantly, NtA does not block elongation, as we demonstrate by visualizing labeled NtA at nascent branch junctions during nucleation/elongation. Rather, NtA stays engaged with Arp2/3 complex and the repeats stay bound to the mother filament after elongation, allowing cortactin to remain tightly bound to the mature branch junction. Our smTIRF measurements are consistent with immuno-gold-labeled electron micrographs and three-dimensional EM reconstructions that show cortactin remains at the mature branch junction (*Cai et al., 2008*;

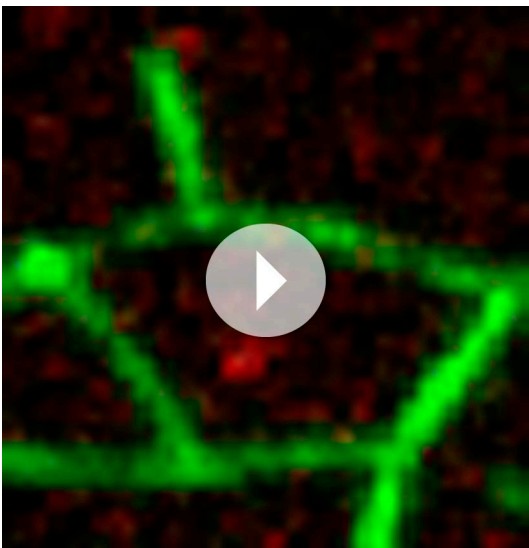

**Video 3**. Single molecules of cortactin binding to preformed branch junctions and filament sides. Preformed branched filament networks were created by polymerizing 1 µM 33% Oregon-Green actin (green), 5 nM Arp2/3 complex and 30 nM VCA for 6.4 min, then flowing into the reaction chamber a solution containing 1.5 nM Alexa568-cortactin (red) and 0.1 µM actin monomers. Out of focus frames (at ~1 s in video) represents when cortactin was flowed into the chamber. Exposure time for each channel was 50 ms with a 561:488 image ratio of 6:1. The frame rate of the 561-channel was 5 fps.

*Egile et al., 2005*). The obligatory displacement model is also consistent with the observation that cortactin competes weakly with GST-VCA in the absence of filaments, but strongly in their presence (*Uruno et al., 2003*).

A key aspect of the obligatory displacement model is that GST-VCA must be released before nucleation. Our mutational analyses support this requirement, since a mutation in the V region that decreased its affinity for actin increased the rate of nucleation at saturating GST-VCA, suggesting weakened interactions with the nascent branch junction can increase nucleation rates. Additional evidence comes from recent single-molecule TIRF experiments, which show that VCA release precedes elongation of the daughter filament (*Martin et al., 2006*) (*Smith et al., 2013*, in press at eLife). Why release of the type I NPF is programmed into the branching nucleation mechanism, at least in the case of dimerized N-WASP-VCA, is not clear. One possibility is that because WASP/Scar proteins are attached to membranes (*Higgs and Pollard, 2000*), incorporation of the release step into the nucleation mechanism ensures branches are not strongly tethered to the membrane as the growing actin network pushes outward on it. In support of this hypothesis, Akin et al. showed that increasing transient connections between bead-immobilized NPFs and nascent branch junctions decreased bead motility (*Akin and Mullins, 2008*). Our data suggest cortactin could stimulate release of a membrane bound NPF from Arp2/3 complex to regulate network-substrate connections, providing an additional mode by which cortactin can control the dynamics of branched actin networks. Recent experiments showed that addition of cortactin to a solution of GST-VCA coated beads increased the bead motility to an extent unlikely to be accounted for simply by an increased nucleation rate (*Siton et al., 2011*). Therefore, cortactin may stimulate branched network dynamics both by increasing nucleation rates and by preventing stalling caused by tight membrane-network attachments.

A second key aspect of the displacement model is that cortactin must be able to target nascent branch junctions and avoid being non-productively sequestered along filament sides. Our single-molecule

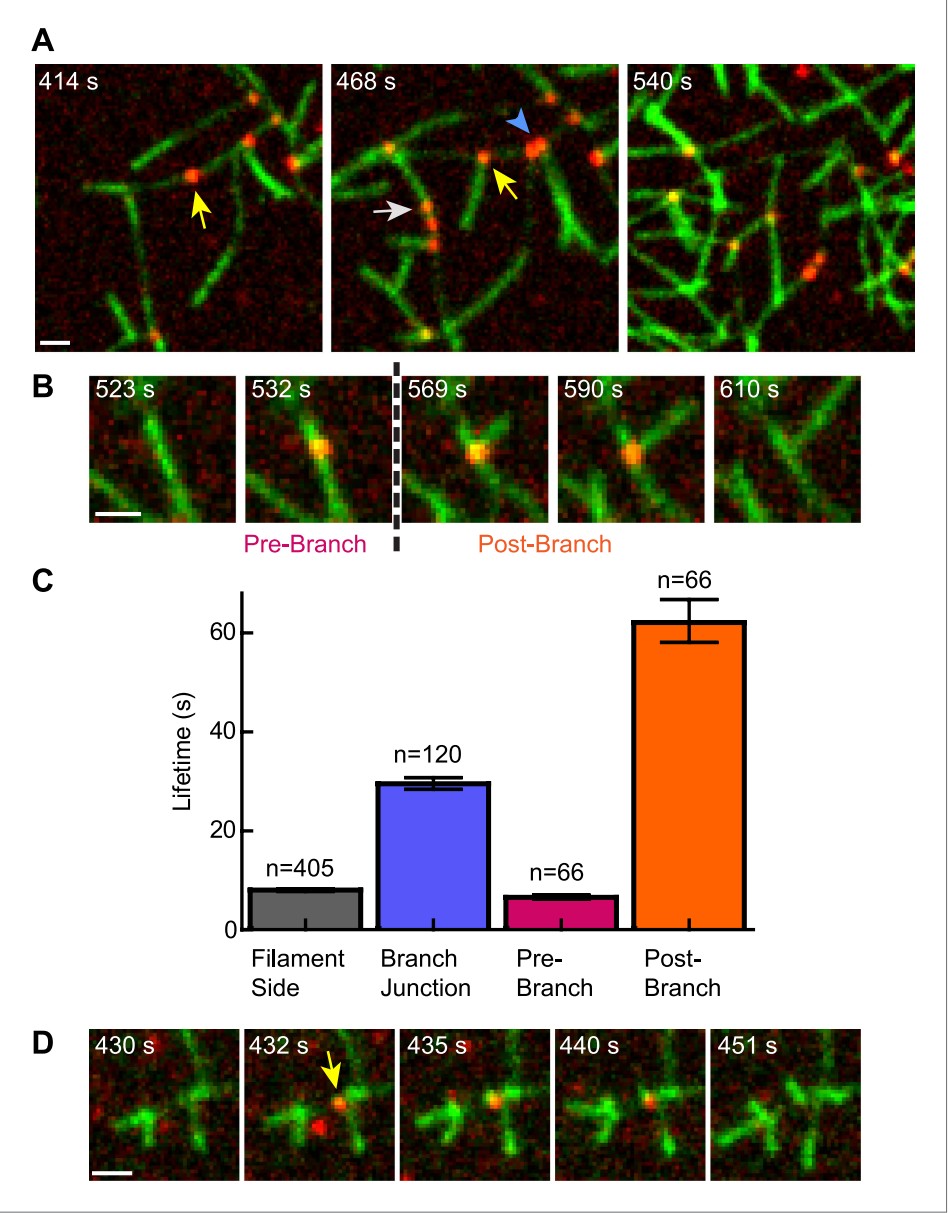

**Figure 8**. Cortactin remains at the branch junction during daughter filament elongation. (**A**) smTIRF microscopy images of polymerizing branch networks containing 1 µM 33% Oregon-Green actin, 5 nM Arp2/3 complex, 50 nM VCA and 2 nM Alexa568-cortactin (red). Images show filament side (gray arrow), existing branch junction (blue arrowhead) and nascent branch junction (yellow arrow) cortactin binding events. (**B**) Montage from reaction described in panel **A** showing a single event in which cortactin binds to a filament side, a new branch is nucleated, and cortactin remains bound during elongation. (**C**) Average lifetimes for each binding class from reaction described in panel **A**. Error bars represent error of the fit. (**D**) Image montage showing NtA (yellow arrow) remains bound for ~4.5 s after daughter filament nucleation before dissociating. The reaction contained 1 µM 33% Oregon-Green actin, 10 nM Arp2/3 complex, 350 nM VCA and 10 nM Alexa568-NtA(1-48). Scale bars: 1 µm.

experiments revealed the kinetic basis for targeting. Instead of diffusing along filament sides to find branch junctions, cortactin targets branches using a ~300-fold increased affinity over filament sides. Most of the binding preference arises from an unusually slow on rate (12,100 $M^{-1}$ $s^{-1}$) of cortactin for filament sides. This on rate is 200–2000-fold slower than for most other actin filament binding proteins, and is unlikely to be diffusion limited (*Kovács et al., 2004*; *De La Cruz et al., 1999*; *Wegner and Ruhnau, 1988*). Binding may be slow because it requires a conformational change in filaments, in

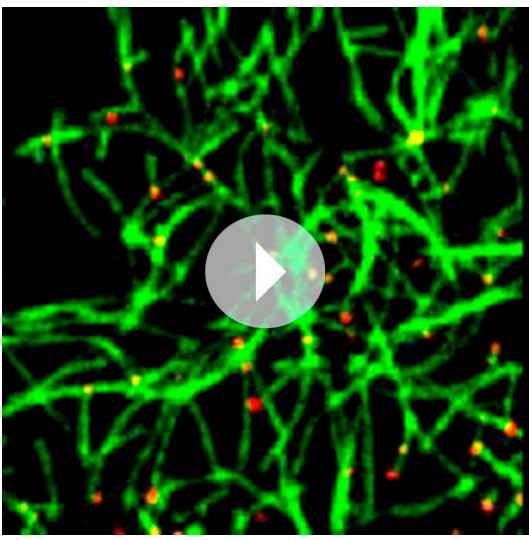

**Video 4**. Single molecules of cortactin binding to branching networks. Reaction contains 1 µM 33% Oregon-Green actin (green), 5 nM Arp2/3 complex, 50 nM VCA and 2 nM Alexa568-cortactin (red). 561- and 488-channel images were exposed for 50 ms and 30 ms, respectively, at a 561:488 image ratio of 8:1, with a frame rate of 5 fps for the 561-channel and 2.6 fps for the 488 channel.

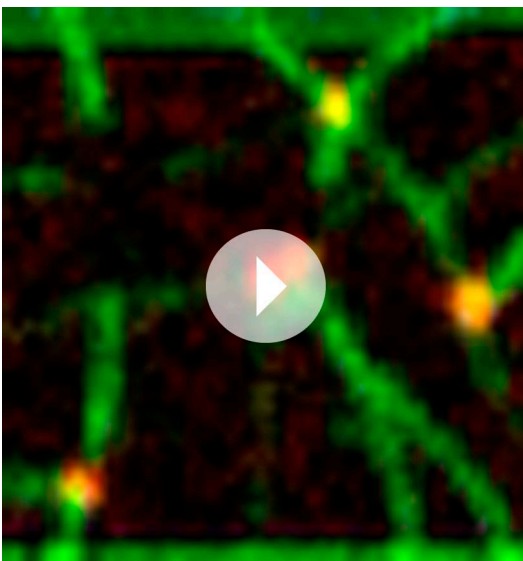

**Video 5**. Single molecule of cortactin binding to a nascent branch junction. Visible in the video are nascent branch (left-center at 3 s), branch junction and filament side binding cortactin molecules. Reaction conditions are the same as *video 4*.

agreement with an electron microscopy reconstruction showing that cortactin binding widens the gap between protofilaments (*Pant et al., 2006*). Our mathematical model of obligatory displacement did not include reactions in which cortactin binds filament sides before Arp2/3 complex rather than targeting an existing nascent branch junction. While these pathways can be inserted into the model, we note that in the case of the NtA construct, synergy must occur completely through nascent branch targeting, suggesting nascent branch targeting may be the dominant displacement pathway for the full-length protein. Additional multi-color smTIRF experiments will be required to fully map out the kinetic pathways.

We showed the actin filament binding repeats of cortactin are not required for synergy, eliminating an actin filament recruitment mechanism. While it is possible that recruitment operates simultaneously with displacement in full-length cortactin, we observed that saturating concentrations of NtA activated nucleation to the same extent as saturating full-length cortactin, so there is no recruitment component of synergy that cannot be mimicked by NtA. The high concentrations of NtA required for saturation likely reflect the decreased affinity of NtA for the nascent branch junction caused by removal of the actin filament binding repeats. Importantly, cortactin can weakly activate Arp2/3 complex on its own (*Weed et al., 2000*), and in contrast to synergy, this intrinsic activity requires the filament binding repeats (*Uruno et al., 2001*). Therefore, filament recruitment may explain the weak intrinsic activity of cortactin observed in vitro. Our data are inconsistent with a previously proposed recycling model of synergy, in which cortactin indirectly increases nucleation rates by recycling WASP sequestered at mature branch junctions, freeing it to activate Arp2/3 complex (*Siton et al., 2011*). Instead, our data indicate that cortactin directly influences nucleation at nascent branch junctions. These biochemical distinctions will have important implications in understanding the influence of cortactin on the regulation of Arp2/3 complex in vivo.

In addition to activating Arp2/3 complex, cortactin has been shown to stabilize branch junctions in vitro (*Weaver et al., 2001*). The average lifetime of cortactin at branch junctions was 29.2 s, whereas the lifetime of branches has been reported to be between 8 and 27 min (*Martin et al., 2006*; *Mahaffy and Pollard, 2006*). Given these data, cortactin dissociates from branches

much more rapidly than branches disassemble. However, cortactin has a high affinity for junctions ($K_D = 17$ nM), so it likely dissociates and rebinds branches many times during the life of a branch, even when present at low (nanomolar) concentrations. Therefore, cortactin dynamically stabilizes branch

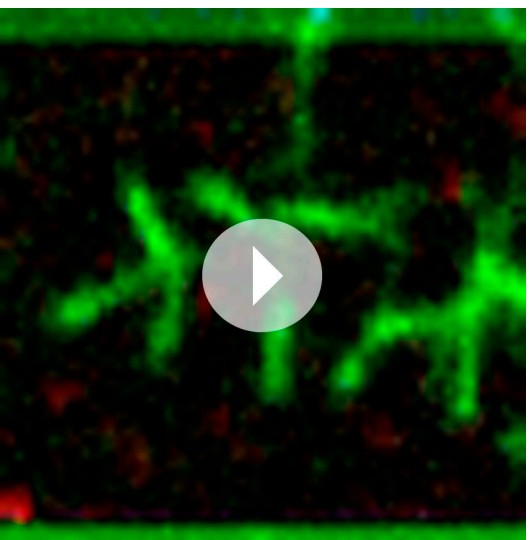

**Video 6**. Single molecule of NtA binding to a nascent branch junction. Reaction contains 1 µM 33% Oregon-Green actin (green), 10 nM Arp2/3 complex, 350 nM VCA and 10 nM Alexa568-NtA(1-48) (red). 561- and 488-channel images were exposed for 50 ms and 30 ms, respectively, at a 561:488 image ratio of 15:1, with a frame rate of 11 fps for the 561 channel and 1.7 fps for the 488-channel.

junctions. This mechanism is consistent with FRAP experiments that show cortactin is incorporated into treadmilling networks not just at the leading edge but also throughout the entire network, and exchanges rapidly (*Lai et al., 2008*). Importantly, in vivo treadmilling networks turn over on much shorter timescales than in vitro branch lifetimes (*Lai et al., 2008*; *Martin et al., 2006*). In cells, the competition of cortactin with branch disassembly factors like GMF, coronin1B, and cofilin may be more important than the intrinsic branch stabilizing activity of cortactin (*Gandhi et al., 2010*; *Cai et al., 2008*; *Blanchoin et al., 2000*).

The mathematical models we present show how rate constants for filament binding by the complex influence the potential of filament recruitment to contribute to activation. Measuring filament on rates has been technically challenging, and a range of values has been reported. Experiments using pyrenyl-Arp2/3 complex from fission yeast yielded a $k_{on}$ of 150 $M^{-1}s^{-1}$ (*Beltzner and Pollard, 2008*), while a $k_{on}$ value of $3 \times 10^3$ $M^{-1}$ $s^{-1}$ was calculated for budding yeast Arp2/3 complex from smTIRF experiments, (*Smith et al., 2013*). Our optimized $k_{on}$ value for bovine Arp2/3 complex was ~$1 \times 10^6$ $M^{-1}s^{-1}$. Species-specific differences may account for some of the differences in these values, but are unlikely to fully explain them. In single-molecule studies, complications arise from the lack of simple methods to directly measure on rates for filaments (*Van Oijen, 2011*). Off rates can generally be determined directly from the lifetimes of the on state, but difficulties arise from the complexity of interactions of Arp2/3 complex with filaments. For example, budding yeast Arp2/3 complex dissociated from filaments with three distinct off rates, indicating a heterogeneous population of dissociating species (*Smith et al., 2013*). Filament binding rate constants are important for not only understanding how Arp2/3 complex works with type I NPFs like WASP/N-WASP, but also how other regulators control Arp2/3 complex activity by mediating its interactions with filaments. The advent of three-color smTIRF experiments will be critical in allowing us to dissect these interactions.

An important finding of this work is that potent synergy occurs only when the type I NPF is dimerized. We hypothesize that this is because dimerized type I NPFs engage both Arp2/3 complex binding sites to bind tightly to nascent branch junctions, thereby slowing the nucleation step. In our assays, we artificially oligomerized N-WASP, but in vivo, N-WASP can oligomerize by association with scaffolding proteins like Nck or BAR domain proteins (*Padrick et al., 2008*; *Li et al., 2012*; *Suetsugu, 2013*). SCAR/WAVE, another widely expressed type I NPF, has also been shown to oligomerize or cluster on membranes, suggesting it can also act as a higher order oligomer (*Gohl et al., 2010*). These observations suggest the potential for cortactin-mediated synergy to activate Arp2/3 complex in multiple distinct branched networks in vivo. We note that in addition to Arp2/3 complex, cortactin has dozens of other binding partners, including some that influence its ability to regulate branched networks (*Kirkbride et al., 2011*). For instance, cortactin binding to WIP, an actin momomer binding protein, greatly enhances cortactin-mediated activation of Arp2/3 complex (*Kinley et al., 2003*). In addition, the SH3 of cortactin binds N-WASP to relieve its autoinhibition, providing an indirect mechanism for cortactin to upregulate branching nucleation (*Kowalski et al., 2005*). Biochemical dissection of these reactions will allow us to understand precisely how cortactin coordinates branched networks in vivo.

## Materials and methods

### Protein purification and labeling

GST-tagged mouse cortactin and cortactin NtA (residues 1–84 or 1–48) (gifts from John Cooper) were overexpressed in BL21(DE3)-RIL *E. coli* and purified using glutathione sepharose, Resource Q ion

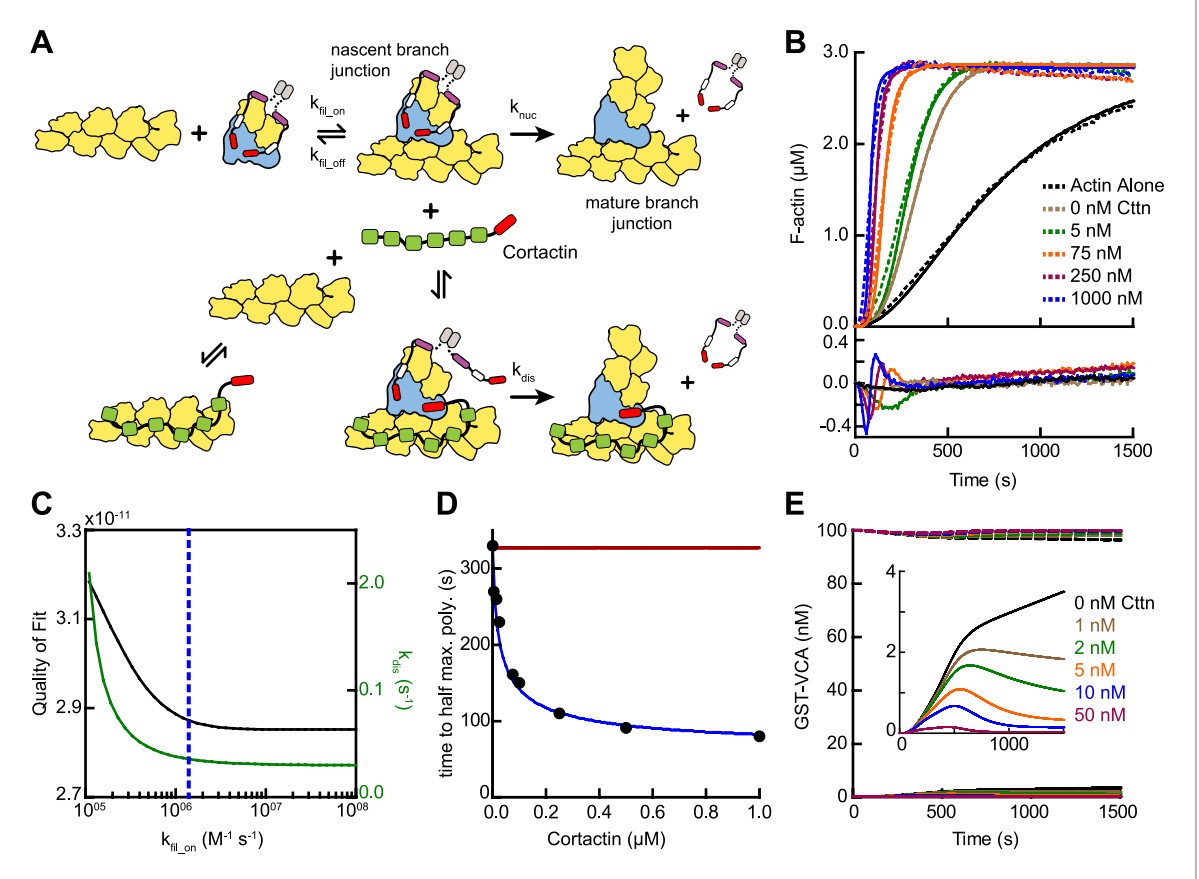

**Figure 9**. Mathematical model of the obligatory displacement mechanism of cortactin-mediated synergy. (**A**) Cartoon pathway of key reactions describing cortactin-mediated displacement of GST-VCA (see 'Materials and methods' for full model). (**B**) Pyrene-actin polymerization time courses of Arp2/3 complex activated by GST-VCA and cortactin with simulated fits based on an obligatory displacement mechanism of synergy. Dashed lines show experimental data and solid lines show simulated fits after optimization of the two floating parameters, $k_{fil\_on}$ and $k_{dis}$. Residuals are shown in lower panel. Reactions contained 3 μM 15% pyrene-actin, 20 nM Arp2/3 complex, 100 nM GST-VCA and indicated concentrations of cortactin. (**C**) Plot of quality of fit (black) and resulting optimized $k_{dis}$ (green) vs $k_{fil\_on}$. Blue dashed line indicates minimum threshold $k_{fil\_on}$ value of $1.4 \times 10^6$ M$^{-1}$ s$^{-1}$. Quality of fit was calculated by a mean-weighted residual sum of squares. (**D**) Plot of half time to equilibrium for reactions in panel **B** (black circles) vs a simulation of the obligatory displacement model (blue line) or the recycling model (red line). (**E**) Plot showing concentration of free GST-VCA (dashed lines, includes any species of GST-VCA not bound to a nascent or mature branch junction) and sequestered GST-VCA (solid lines, includes any species bound at nascent or mature branch junction) vs time in simulations of the recycling model run at a range of cortactin concentrations. Insert shows magnification of a section of the plot, highlighting the concentration of sequestered GST-VCA as a function of cortactin concentration.

exchange and size exclusion chromatography columns, in that order. The GST tag was cleaved using TEV protease prior to ion exchange chromatography. Cortactin (residues 1–336) or NtA (1–48) used in smTIRF were prepared for dye labeling by mutating all endogenous cysteines to serine and adding a KCK (Lys-Cys-Lys) tag at the C-terminus. Purified fractions from size exclusion chromatography were reacted with Alexa Fluor 568 maleimide (Molecular Probes, Eugene, OR) at a molar ratio of 15–20:1 (dye:cortactin) overnight at 4°C. Free dye was removed by extensive dialysis and a HiTrap desalting column. The concentration of labeled cortactin (1–336) was determined using absorbance at 280 nm with the dye signal subtracted using the 280:575 nm absorbance ratio of free dye. Labeled NtA concentration was estimated by measuring the dye absorbance at 575 nm and assuming that all free dye was removed and 100% of the protein was labeled. LZ-VCA was constructed by inserting the leucine zipper domain of *Saccharomyces cerevisiae* Gcn4 (residues 250-281, a gift from Alan Hinnebush) and a 21 residue Gly-Ser linker between the TEV protease site and the N-terminus of VCA. All N-WASP-VCA constructs, including N-WASP-VVCA (residues 392–505), were purified as described previously (*Hetrick et al., 2013*). Bovine Arp2/3 complex, Oregon-Green actin, rabbit muscle actin and pyrene-labeled actin were prepared as previously described (*Hetrick et al., 2013*).

## Pyrene actin polymerization assays

Pyrene actin polymerization assays were performed and analyzed as described previously (*Liu et al., 2011*). Fold activation due to cortactin-mediated synergy was calculated by dividing the maximum polymerization rate at a given cortactin concentration by the maximum polymerization rate of the equivalent reaction containing no cortactin. Synergistic activation vs cortactin concentration data were fit to a saturating hyperbolic equation with an added background factor equal to the activity at no cortactin.

## Competition binding assays

The anisotropy of a solution containing 50 nM Rhodamine-VCA, 150 nM G-actin, 10 mM Hepes pH 7.0, 50 mM KCl, 1 mM $MgCl_2$, 1 mM EGTA, 1 mM DTT, 0.2 mM ATP, 350 nM Latrunculin B and increasing concentrations of VCA or VCA mutants was measured using excitation and emission wavelengths of 530-nm and 574-nm, respectively. Plots of anisotropy vs VCA concentration were fit with a previously described equation (*Wang, 1995*). Binding assays were repeated at least three times. The final reported $K_D$s are the mean of the individual fits.

## TIRF buffers

The TIRF reaction buffer contained 50 mM KCl, 1 mM $MgCl_2$, 1 mM EGTA, 10 mM Imidazole pH 7.0, 0.5% methyl cellulose cP 400, 1 mM DTT, 0.2 mM ATP, 25 mM glucose, 1 mM Trolox, 0.02 mg/mL catalase, 0.1 mg/mL glucose oxidase and 2 mg/mL BSA. Methyl cellulose (2% cP 400) stock was prepared by overnight mixing at 4°C and then centrifuged for 1 hr at 245,070×*g* in a table-top ultracentrifuge. A stock solution of 5 mg/ml glucose oxidase and 1 mg/ml catalase (GODCAT) was prepared according to manufacturer's instructions in water and stored at −20°C. A 2× stock of TIRF buffer was prepared by mixing KMEI (50 mM KCl, 1 mM $MgCl_2$, 1 mM EGTA, 10 mM Imidazole pH 7.0), glucose and Trolox and incubating overnight at room temperature to allow for formation of the Trolox quinone derivative (*Cordes et al., 2009*). ATP, DTT and the clarified methyl cellulose were added before filtering the mixture through a 0.22 µm filter and storing at −20°C. Just before use, GODCAT and BSA were added to make the final 2× TIRF buffer stock.

## PEGylated TIRF chamber assembly and reaction preparation

Cover slips (no. 1.5) were sonicated for 20–25 min in acetone, then 1 M KOH with extensive water rinsing between sonication steps. Cover slips were then briefly washed in methanol before incubating for 30 min in a 1% APTES ((3-Aminopropyl)triethoxysilane), 5% acetic acid methanol solution. Aminosilanized cover slips were rinsed with methanol then water and allowed to completely dry. TIRF chambers were assembled by adhering two strips of double-sided tape, separated by 5 mm, along the long axis of an aminosilanized cover slip. A 24 mm wide, 1 mm thick microscope slide was rinsed briefly with ethanol and placed perpendicular across the cover slip such that the middle of the slide and cover slip were aligned. Firm, even pressure was applied to the slide to secure it to the cover slip tape. The resulting assembly contains a 5 mm × 24 mm × 0.1 mm (tape height) reaction chamber with openings on two sides for reaction solution addition and exchange. Solution exchange was performed by using Whatman paper to wick out the chamber solution from one end resulting in the uptake of the new solution into the chamber, from the opposite end.

TIRF reaction chambers were then treated with biotinylated-PEG to create a low binding surface to which actin filaments could be tethered through streptavadin and biotinylated-myosin. A single-use solution of 250–300 mg/mL methoxy PEG succinimidyl succinate (JenKem USA, Allen, TX) ~0.1% Biotin PEG NHS Ester was prepared in 0.22 µm filtered 0.1 M $NaHCO_3$ pH 8.3. TIRF chambers were prepared for PEGylation by flowing in 0.1 M $NaHCO_3$ pH 8.3 twice. The PEG solution was then wicked into the reaction chamber and allowed to incubate for 4–5 hr at room temperature, protected from light and in a humidity chamber to prevent the reaction chamber from drying. After PEGylation, filtered water was flowed through the reaction chamber 3–5 times to remove all the unbound PEGs. PEGylated chambers were stored at 4°C in a light protected humidity chamber for up to 1 week.

Immediately before imaging, individual TIRF reaction chamber surfaces were prepared by flowing in 50–100 µM streptavidin followed by 0.5 µM biotinylated myosin, which was previously prepared by reacting maleimide-biotin with full length rabbit myosin II (Cytoskeleton Inc., Denver, CO) on ice for 2–4 hr and stored at 4°C. The chamber was then washed with high then low salt buffer solutions (20 mg/mL BSA, 50 mM Tris pH 7.5 and 600/150 mM NaCl, respectively), followed by two washes with

1× TIRF buffer with GODCAT and BSA (see above). Each solution was allowed to incubate in the TIRF chamber for 5-10 min. TIRF actin polymerization reactions were initiated by adding a protein solution containing the protein(s) of interest (Arp2/3 complex, Alexa568-cortactin, VCA) in 1× TIRF buffer to a solution of 6 μM 33% Oregon-Green actin, pretreated for 60 s with 50 μM $MgCl_2$ and 200 μM EGTA, to give a final reaction solution of 1 μM 33% Oregon-Green actin and the correct concentration of the proteins of interest (see below). The reaction was then wicked into the prepared reaction chamber and imaged as described below.

## smTIRF microscopy

Dual wavelength TIRF images were collected on a Nikon TE2000-U microscope outfitted with a Nikon 100× NA 1.49 TIRF objective, 1.5× auxiliary lens and an EM-CCD camera (iXon3, Andor or Image-EM, Hamamatsu). Argon 488-nm (Dynamic Laser, Salt Lake City, UT) and solid-state 561 nm (Coherent, Santa Clara, CA) lasers were used to excite Oregon-Green and Alexa-568 fluorophores, respectively. Laser beam selection and intensity was controlled using an AOTF (Gooch & Housego) and each beam passed through dual-band (488/561) excitation and dichroic filters (Chroma, Bellows Falls, VT) before entering the objective. Prior to collection at the EM-CCD, emission signals passed through dual-band dichroic and emission filters (Chroma). Images were acquired using the open source microscopy software, Micro-Manager (*Edelstein et al., 2010*). Image processing was performed in ImageJ, where each raw image was background subtracted using the rolling ball algorithm with a rolling ball radius of 10 pixels and subsequently smoothed using a Gaussian blur filter with a sigma of 0.5. Unless noted otherwise, TIRF reactions were imaged at a final magnification of 150× with 50 ms and 30–50 ms 561- and 488-channel exposures, respectively. Images were acquired at 561:488 image channel ratios of 5–15:1 and at calculated frame rates of 1–11 frames per second. Specific imaging parameters are indicated in figure or video legends.

## Lifetime analysis

Single molecules were tracked and lifetimes measured using a custom Matlab script (MathWorks, Natick, MA). Single cortactin molecules were identified in each 561-channel image using thresholding image segmentation after removal of noise using a band pass filter. Single-molecule trajectories were created by identifying and linking identical molecules between frames using a nearest neighbor algorithm. An assembled trajectory represents a single-molecule and contains a preliminary frame-based lifetime. All identified molecules were filtered based on the following criteria: Molecule cannot be present in the first or last frame of the image acquisition period, molecule cannot have a lifetime of 1 frame, molecule average intensity must be within a standard deviation of the overall molecular average intensity and the molecule must be associated with an actin filament based on filament identification statistics performed on the corresponding 488-channel images. The molecules that passed the filter were manually tracked to verify the initially determined frame lifetime and to identify the binding class (filament side, branch junction, nascent branch). We identified branches manually by visually inspecting videos. Potential branches were examined in multiple frames to verify that they were not overlapping linear actin filaments. Lifetimes were converted from frames to seconds based on calculated frame rates from image time stamps, and binned into 5 s intervals. The cumulative frequency across all bins was calculated and a plot of 1- cumulative frequency vs lifetime (bin) was fit with a single-exponential decay equation to determine the average off rate from which the average lifetime is the reciprocal.

## Single-molecule affinity determination

Single-molecules of cortactin were identified (see above) and classified into filament side or branch junction binding from 1000 randomly chosen frames of the preformed branch network image acquisition. Start of equilibrium binding was determined from a plot of number of particles vs frame. Number of filament binding sites was calculated for each frame at equilibrium (829 frames) by a custom image processing script run in Matlab, described as follows. For each frame, pixels corresponding to filament fluorescence were identified using image segmentation followed by morphological area opening to remove non-filamentous small fluorescent objects. Pixels corresponding to branch junctions (5 pixels per junction) were subtracted from the total number of pixels and this new number of pixels was divided by 3 (average filament width in pixels) to remove pixels corresponding to the PSF of the filament fluorescence. The final pixel number value was converted to micrometers (1 px = 106.7 nm) to yield the total length of actin filaments in the image frame, and further converted to number of

subunits using 370 subunits μm⁻¹ (***Kuhn and Pollard, 2005***). A 4% error in total filament length was found between using the above algorithm and manual filament tracing on a small subsection of filaments; therefore, the described length calculation algorithm works well for the extensive length measurements needed. The total number of cortactin filament side binding sites was calculated by assuming a cortactin to F-actin subunit stoichiometry of 1:6. The total number of branch junctions was visually counted in each frame. For each frame, the fraction bound of actin for each binding group was calculated by dividing the total number of counted molecules by the total number of binding sites. The affinity for each binding group was calculated, assuming excess ligand conditions (see below), using the equation: $Kd = \dfrac{[cortactin]}{fraction\ bound} - [cortactin]$, where fraction bound is the average across all analyzed frames and concentration of cortactin is 1.5 nM.

Excess ligand conditions were established using the following calculations. The average area of a reaction chamber is 120 mm² and the area of a single image is $2.98 \times 10^{-3}$ mm² (512 × 512 pixels at 106.7 nm per pixel), indicating that, in two dimensions, a single image composes 0.00248% of the total chamber area. The average number of cortactin molecules bound to junctions or filaments sides per image frame was 20. Because unbound actin filaments were washed out of the chamber, all binding events occurred on the surface, so this number allows us to account for all actin-bound cortactin molecules in the chamber. Using the chamber-to-image ratio ($2.48 \times 10^{-5}$), this gives an average number of cortactin molecules bound per chamber of $8.06 \times 10^{5}$. In the reaction chamber there are $1.08 \times 10^{10}$ cortactin molecules (12 μl of 1.5 nM cortactin) indicating that on average 0.0075% ($8.06 \times 10^{5}/1.08 \times 10^{10}$) of the total cortactin molecules are bound to actin, therefore the free cortactin concentration is essentially equal to its total concentration.

## Mathematical modeling of Arp2/3 complex nucleation by GST-VCA activation with and without cortactin

To limit the number of floating parameters in our kinetic models, we ran four sets of pyrene actin polymerization assays, and fit each set to an independent model with a limited number of floating variables (***Table 2***). For example, to determine rate constants for spontaneous actin filament nucleation, we fit time courses of a range of concentrations of actin polymerizing without additional proteins (***Figure 3—figure supplement 1A,B***). The optimized rate constants from this model were used to fit sets of pyrene actin polymerization assays containing additional proteins (GST-VCA, Arp2/3 complex, and cortactin, i.e., reaction sets 1–4). To determine a set of reactions and rate constants that can describe decreased nucleation from GST-VCA-bound actin monomers, we modeled a set of reactions containing a constant concentration of actin and a range of GST-VCA concentrations (***Figure 3—figure supplement 1A,C***). ***Table 2*** shows each of the four sets of reactions and the simulations of their optimized global fits.

While many of the rate constants for interactions in the branching nucleation reaction are known (***Tables 1 and 2***), we made several assumptions to allow construction of the model. Rate constants of GST-VCA binding to actin monomers were assumed to be the same as monomeric VCA (***Marchand et al., 2001***). Kinetic rate constants for GST-VCA binding to Arp2/3 complex have not been measured, so we used $k_{on}$ values measured for monomeric VCA and adjusted the $k_{off}$ to account for the previously measured tighter binding of GST-VCA to Arp2/3 complex (***Padrick et al., 2008***). We decreased the $k_{on}$ of actin for GST-VCA bound to Arp2/3 complex and increased the $k_{off}$ of GST-VCA:actin from Arp2/3 complex to account for competition between the complex and C for actin binding (***Kelly et al., 2006***). We modeled the Arp2/3 complex activating nucleation step ($k_{nuc}$) by converting the nascent branch junction of two actin monomers bound to GST-VCA bound to Arp2/3 complex at a filament side to a barbed end which subsequently elongated at $11.6 \times 10^{6}$ M⁻¹ s⁻¹ (***Pollard, 1986***). Pointed end elongation was not included in our simulations, because the actin monomer concentration is low and Arp2/3 complex-mediated nucleation does not create free pointed ends. We assumed GST-VCA dissociates from the branch junction during nucleation ($k_{nuc}$), except in the recycling model (see main text and ***Figure 3—figure supplement 1A***). For simplicity, the $k_{on}$ of Arp2/3 complex for the sides of filaments was assumed to be unaffected by GST-VCA or GST-VCA and actin monomer binding. For reactions with cortactin, the stoichiometry of cortactin:F-actin subunits was set to 1:6. Rate constants for cortactin binding to filament sides and branch junctions were determined from the single-molecules studies.

Mathematical modeling of pyrene-actin polymerization time courses was performed using COPASI (***Hoops et al., 2006***). Fluorescence values were converted to actin filament concentrations by assuming

0.1 µM actin was unpolymerized at equilibrium. Optimization of parameters was carried out by simultaneously fitting all traces from a reaction set, using the Levenberg–Marquardt algorithm method in the parameter estimation module. To simulate the influence of actin filament recruitment by cortactin (*Figure 3D*), we increased the initial concentration of actin filament sides but not ends.

## Acknowledgements

We would like to thank Bruce Bowerman and Chris Doe for use of their microscope, Raghuveer Parthasarathy for the particle tracking script, and members of the BJN laboratory for helpful discussions. We also thank Shae Padrick, Mike Rosen and Ben Smith for discussing unpublished observations, Julien Berro for advice on mathematical models, and Jon Cooper and Alan Hinnebush for sending plasmids.

## Additional information

### Funding

| Funder | Grant reference number | Author |
|---|---|---|
| National Institutes of Health | R01-GM092917 | Luke A Helgeson |
| Pew Biomedical Scholar Program | | Luke A Helgeson |
| National Institutes of Health Predoctoral Training Grant | GM007759 | Luke A Helgeson |

The funders had no role in study design, data collection and interpretation, or the decision to submit the work for publication.

### Author contributions

LAH, Conception and design, Acquisition of data, Analysis and interpretation of data, Drafting or revising the article; BJN, Conception and design, Analysis and interpretation of data, Drafting or revising the article

## Additional files

### Major datasets

The following previously published datasets were used:

| Author(s) | Year | Dataset title | Dataset ID and/or URL | Database, license, and accessibility information |
|---|---|---|---|---|
| Chereau D, Kerff F, Graceffa P, Grabarek Z, Langsetmo K, Dominguez R | 2005 | Ternary complex of the WH2 Domain of WIP with Actin-DNAse I | 2A41; http://www.rcsb.org/pdb/explore/explore.do?structureId=2A41 | Publicly available at the RCSB Protein Data Bank (www.rcsb.org/pdb/). |
| Gaucher JF, Mauge C, Didry D, Guichard B, Renault L, Carlier MF | 2012 | Crystal structure of N-Wasp VC domain in complex with skeletal actin | 2VCP; http://www.rcsb.org/pdb/explore/explore.do?structureId=2VCP | Publicly available at the RCSB Protein Data Bank (www.rcsb.org/pdb/). |

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
