## [Decision Letter]

Thank you for sending your work entitled “Mechanism of synergistic activation of Arp2/3 complex by cortactin and N-WASP” for consideration at *eLife*. Your article has been favorably evaluated by a Senior editor and 3 reviewers, one of whom, Wesley Sundquist, is a member of our Board of Reviewing Editors.

The Reviewing editor and the other reviewers discussed their comments before we reached this decision, and the Reviewing editor has assembled the following comments to help you prepare a revised submission.

The authors present experiments and simulations aimed at explaining the mechanism by which cortactin (and its minimal NtA domain) synergizes with WASP (and its minimal VCA activation region) to stimulate actin branching. They conclude that displacement of dimeric WASP(VCA) is limiting for branch formation, and that cortactin accelerates the branching reaction primarily by displacing WASP(VCA). In support of this model, they show that: 1) actin filament recruitment cannot account for cortactin activity. Here, the strongest experiment is the demonstration that actin filament binding activity is not required for maximal cortactin stimulation because the minimal NtA domain (which lacks the actin filament binding repeats) stimulates to the same extent, at saturation, as full length cortactin; 2) VCA dimerization is required for cortactin stimulation and, nicely, mutations that alter the rate of N-WASP dissociation alter actin polymerization (and the degree of cortactin stimulation) as predicted by the model (Figure 3); 3) cortactin binds preferentially to branch sites because it associates much faster onto branch sites than onto the sides of actin filaments, and also dissociates a bit slower (direct visualization/binding studies, Figures 4 and 5), and can remain there as the actin filament nucleates and elongates (Figure 6, lifetime measurements); and 4) simulations reveal that data for GST-VCA and cortactin co-stimulation of actin polymerization can be explained by a displacement model, but not by a VCA recycling vs sequestration model (because GST-VCA binding does not significantly deplete the pool of free GST-VCA).

This is generally a very nice study that addresses an important problem in actin biology – that is, how different actin nucleating factors act together to regulate actin assembly. Overall the conclusions are well supported by the data presented.

No additional experiments are absolutely required, but the authors need to discuss the following issues more explicitly:

1) Why cortactin synergizes with dimeric VCA but not monomeric VCA. The authors argue that the difference between dimeric vs monomeric VCA data is that the off rate of monomeric VCA is sufficiently high that cortactin, at the concentrations explored, does not play a role in displacing it. Does the apparent lack of synergy for monomeric VCA depend on VCA concentration or on Arp2/3 concentration? It might also be useful to extend the study to explore a broader range of concentrations. More importantly, the dimeric VCA formed with GST may have an artificially low off rate from the branch junction compared to two monomeric VCAs on a branch junction, making the observed ‘displacement’ relevant to the dimeric VCA construct but not relevant in vivo. The increased activation by monomeric VCA at saturation compared to dimeric VCA underscores this issue (Figure 3). Can the authors address this concern?

2) The role of the filament binding domain of cortactin in targeting branch junctions would benefit from further clarification. How does the on rate and off rate of NtA at branch junctions (imaged in Figure 6) compare with full-length cortactin rates presented in Figure 5?

3) The study would also benefit from some discussion of how nucleotide state of the filament affects cortactin binding to branch junctions.

4) Finally, the authors should exert some effort to make their paper, particularly the Introduction, more accessible to the general readers of *eLife*. The authors are urged to extend their reach to an audience outside the actin polymerization field. (i) Consider providing an introductory set of schematic diagrams at the very beginning that brings together parts of the schematics scattered throughout the figures so that the reader is introduced to the players and can better understand what the issues are, when reading the Introduction. (ii) Consider reducing the extent to which three letter acronyms are used – this makes the paper rather impenetrable to the outsider. For example, why not spell out nucleation promoting factor rather than resorting to the opaque “NPF”? (iii) Explain the conceptual problem that is being addressed in way that can be better understood by outsiders. For example, in the very last part of the Introduction, two prior models are described, the second of which is that the role of cortactin is to “stabilize the nucleus or recycle VCA back into solution”. The authors then go on to say that they have a different model, referred to as a “displacement” model. It is not clear what the distinction is, at least when reading the Introduction.

---

## [Author Response]

*1) Why cortactin synergizes with dimeric VCA but not monomeric VCA. The authors argue that the difference between dimeric vs monomeric VCA data is that the off rate of monomeric VCA is sufficiently high that cortactin, at the concentrations explored, does not play a role in displacing it. Does the apparent lack of synergy for monomeric VCA depend on VCA concentration or on Arp2/3 concentration? It might also be useful to extend the study to explore a broader range of concentrations*.

Synergy between monomeric VCA and cortactin is consistently weak over a range of concentrations of monomeric VCA. Figure 4 shows that the increase in the maximum polymerization rate of reactions with either 150 or 750 nM monomeric VCA at saturating cortactin is 1.5 fold, compared to 3.7x for dimeric VCAs. These concentrations of monomeric VCA were chosen based on the concentration dependence plots shown in Figure 5, with one sub-saturating concentration (150 nM) and one concentration at which the reaction is saturated with monomeric VCA (750 nM). In a separate set of experiments, we titrated 500 nM cortactin and 20 nM Arp2/3 complex with monomeric VCA, and showed that the fold activation (synergy) is consistently ∼1.5x at monomeric VCA concentrations ranging from 0.25 to 4.0 μM. We included this new data as Figure 4 and added text to the Results section (starting “Saturating corcactic enhanced the maximum polymerization rate of a reaction…”).

We have not tested synergy over a broad range of Arp2/3 complex concentrations. By keeping the Arp2/3 complex concentrations low in our assays, we avoid having to account for a small amount of direct activation of the complex by cortactin, which could complicate the analysis.

*More importantly, the dimeric VCA formed with GST may have an artificially low off rate from the branch junction compared to two monomeric VCAs on a branch junction, making the observed ‘displacement’ relevant to the dimeric VCA construct but not relevant in vivo. The increased activation by monomeric VCA at saturation compared to dimeric VCA underscores this issue (*Figure 3*). Can the authors address this concern*?

We predict the off rate of any dimeric VCA from the nascent branch will be slowed compared to monomeric VCAs because the dimers can make multivalent interactions with the complex. In support of this, we showed that VCA dimerized by a leucine zipper shows the same magnitude of synergy with cortactin as GST-VCA. While the leucine zipper and GST are artificial dimerizers, a number of SH3 domain-containing proteins can bind the polyproline stretches in WASP/Scar family proteins to create oligomers which activate Arp2/3 complex similarly to GST-VCA (43). We are currently generating reagents to test whether WASP dimerized in this more physiologically relevant manner shows the same potency in synergy as GST-VCA, but we have no reason to think it will not. One possibility is that the length of the linker between the V region and the dimerization domain could influence synergy. In our GST-VCA construct, the linker between GST and V is 16 amino acids long, similar to the distance between the polyproline stretch and the first V in full-length N-WASP (∼16 amino acids) and the V in WASP (∼13 amino acids), or Scar/WAVE (∼22 amino acids). The linker is longer in our LZ-VCA construct (27 amino acids), which behaves identically to GST-VCA. Together these observations suggest the behavior of the dimers is consistent within the range of linker lengths we expect for physiological dimers.

Because GST-dimerized VCA behaves similarly to VCA dimerized through more physiological mechanisms, we think the decreased activity of GST-VCA compared at saturation to monomeric VCA will be relevant in vivo. To our knowledge, there are few experiments that directly address the oligomerization state of these NPFs in a cell during Arp2/3 complex-dependent actin assembly (23). However, given the biochemical data it seems likely that many of these actin assembly processes depend on dimerization or higher order oligomerization of the NPF. Therefore the activity profile of GST-VCA in Figure 5 (and its associated kinetic parameters) may be equally (or more) relevant in dissecting the kinetics of branched network assembly in cells than kinetic parameters for monomeric VCA.

We expanded the Results section to clarify this issue (sentences starting “Therefore, we compared cortactin-mediated synergy…”).

*2) The role of the filament-binding domain of cortactin in targeting branch junctions would benefit from further clarification. How does the on rate and off rate of NtA at branch junctions (imaged in*
Figure 6*) compare with full-length cortactin rates presented in*
Figure 5?

Unfortunately, we did not observe enough NtA binding events to calculate the binding rate constants. This is due to the weaker affinity of NtA for the branch junctions compared to full-length cortactin and an upper threshold on the concentration of labeled NtA we could use in the reaction.

*3) The study would also benefit from some discussion of how nucleotide state of the filament affects cortactin binding to branch junctions*.

We analyzed the ages of actin filament segments in movies of cortactin binding to both preformed and actively assembling branches (Video 3 versus Video 4, respectively). In the movies with preformed branches, most actin subunits within the filament segments are bound to ADP, whereas in the actively branching movies, most actin subunits are ADP-Pi bound. Despite this difference, the average lifetime of cortactin bound at branch junctions was the same in two sets of movies (∼29 s). Therefore, we don’t think that the nucleotide state of the filament, at least comparing ADP to ADP-Pi bound actin, has a significant influence on cortactin binding to branch junctions.

*4) Finally, the authors should exert some effort to make their paper, particularly the introduction, more accessible to the general readers of* eLife*. The authors are urged to extend their reach to an audience outside the actin polymerization field. (i) Consider providing an introductory set of schematic diagrams at the very beginning that brings together parts of the schematics scattered throughout the figures so that the reader is introduced to the players and can better understand what the issues are, when reading the Introduction*.

We thank the reviewers for this suggestion. We added a new figure (Figure 1) with three panels. The first panel, (A), shows an overview of the branching nucleation reaction, emphasizing what is required for the reaction and illustrating the resultant Y-shaped branches. The second panel, (B), shows the domain organization of N-WASP, WASP and cortactin. The third panel, (C), shows an overview of activation of Arp2/3 complex by Type I and Type II NPFs alone or in synergy, emphasizing the interactions and issues that are addressed in the manuscript.

*(ii) Consider reducing the extent to which three letter acronyms are used – this makes the paper rather impenetrable to the outsider. For example, why not spell out nucleation promoting factor rather than resorting to the opaque “NPF”*?

We understand the downsides to including acronyms but after careful consideration, we hope the following changes will be accepted as a substitute for elimination of the acronyms we use in the manuscript:

1. We explicitly defined the acronyms WASP, Scar, and Arp2/3 complex at first mention.

2. We added Figure 1, which reinforce the meanings of many of the acronyms used in the manuscript with visual representations.

3. We replaced “G-actin” with “actin monomers” and “F-actin” with “actin filaments” wherever possible.

*(iii) Explain the conceptual problem that is being addressed in way that can be better understood by outsiders. For example, in the very last part of the Introduction, two prior models are described, the second of which is that the role of cortactin is to “stabilize the nucleus or recycle VCA back into solution”. The authors then go on to say that they have a different model, referred to as a “displacement” model. It is not clear what the distinction is, at least when reading the Introduction*.

To address this issue we added Figure 1, which we hope gives the general reader a better understanding of the problems being addressed. In addition, we changed the name of our model from “displacement” to “obligatory displacement” to clearly distinguish it from other models. We also revised the Introduction (sentences starting “Previously proposed models include scenarios…”).